DOI: 10.1038/s41467-017-02188-7　　**OPEN**

# Integrative transcriptomic analysis reveals key drivers of acute peanut allergic reactions

C.T. Watson[1,2], A.T. Cohain[1,3], R.S. Griffin[4], Y. Chun[1], A. Grishin[5], H. Hacyznska[1], G.E. Hoffman [1,3], N.D. Beckmann[1,3], H. Shah[1], P. Dawson[6], A. Henning[6], R. Wood[7], A.W. Burks[8], S.M. Jones[9], D.Y.M. Leung[10], S. Sicherer[5], H.A. Sampson[5], A.J. Sharp[1], E.E. Schadt [1,3] & S. Bunyavanich[1,5]

Mechanisms driving acute food allergic reactions have not been fully characterized. We profile the dynamic transcriptome of acute peanut allergic reactions using serial peripheral blood samples obtained from 19 children before, during, and after randomized, double-blind, placebo-controlled oral challenges to peanut. We identify genes with changes in expression triggered by peanut, but not placebo, during acute peanut allergic reactions. Network analysis reveals that these genes comprise coexpression networks for acute-phase response and pro-inflammatory processes. Key driver analysis identifies six genes (*LTB4R*, *PADI4*, *IL1R2*, *PPP1R3D*, *KLHL2*, and *ECHDC3*) predicted to causally modulate the state of coregulated networks in response to peanut. Leukocyte deconvolution analysis identifies changes in neutrophil, naive CD4[+] T cell, and macrophage populations during peanut challenge. Analyses in 21 additional peanut allergic subjects replicate major findings. These results highlight key genes, biological processes, and cell types that can be targeted for mechanistic study and therapeutic targeting of peanut allergy.

[1] Department of Genetics and Genomic Sciences, Icahn School of Medicine at Mount Sinai, New York, NY 10029, USA. [2] Department of Biochemistry and Molecular Genetics, University of Louisville School of Medicine, Louisville, KY 40202, USA. [3] Icahn Institute for Genomics and Multiscale Biology, Icahn School of Medicine at Mount Sinai, New York, NY 10029, USA. [4] Department of Anesthesia, Hospital for Special Surgery, New York, NY 10021, USA. [5] Department of Pediatrics, Icahn School of Medicine at Mount Sinai, New York, NY 10029, USA. [6] eEmmes Corporation, Rockville, MD 20850, USA. [7] Department of Pediatrics, Johns Hopkins University, Baltimore, MD 21287, USA. [8] Department of Pediatrics, University of North Carolina, Chapel Hill, NC 27599, USA. [9] Department of Pediatrics, University of Arkansas for Medical Sciences and Arkansas Children's Hospital, Little Rock, AS 72202, USA. [10] Department of Pediatrics, National Jewish Health, Denver, CO 80206, USA. C.T. Watson and A.T. Cohain contributed equally to this work. E.E. Schadt and S. Bunyavanich jointly supervised this work. Correspondence and requests for materials should be addressed to S.B. (email: supinda@post.harvard.edu)

Peanut allergy is a clinical and public health problem that affects 2–5% of US school-age children, with evidence of increasing prevalence[1]. Peanut allergy is defined as an adverse health effect arising from a specific immune response that occurs reproducibly upon exposure to peanut, leading to acute symptoms such as hives, swelling, respiratory difficulty, cardiovascular compromise, gastrointestinal disturbance, and/or anaphylaxis that can be life-threatening[2]. The development of peanut allergy involves deviation from mucosal and cutaneous immune tolerance, such that dietary antigens presented by antigen presenting cells lead to an adverse Th2-cell skewed response, priming of innate immune effector cells, and alteration of cytokine milieu such that subsequent food allergen-specific antigen exposure leads to IgE-mediated acute reactions[3]. In addition to immediate symptoms of the allergic response, peanut allergy has broader long-term effects, impacting nutrition, emotional health, and lifestyle[2]. There is presently no cure for peanut allergy, and our understanding of its causation and pathobiology remain limited.

Gene expression studies in food allergic individuals have provided some insight into the potential molecular processes underlying peanut allergy. Differentially expressed genes identified in food allergic individuals indicate that underlying perturbations in immune system processes contribute to the development of food allergy[4, 5]. For example, transcriptome-wide gene expression profiles in stimulated CD4+ T cells collected from allergic individuals at infancy prior to disease onset, and matched non-allergic controls, revealed significant expression differences for >2000 genes, which together represented a signature of reduced capacity for T cell proliferation in allergic individuals[4]. Comparisons of gene expression levels in circulating peripheral blood mononuclear cells between 17 fruit and/or latex allergic adults and four controls revealed global differences in the resting state, including the differential expression of genes involved in helper T cell cytokine signaling[5]. Dysregulation of genes involved in various immune processes has also been reported in other allergic diseases[6, 7].

Although studies conducted to date have illuminated factors and genes associated with food allergy susceptibility, little emphasis has been placed on characterizing genes that mediate acute food allergic reactions. Furthermore, studies attempting to address such questions have not done so in human cohorts using in vivo oral food challenges. Thus, limited actionable information exists that can be used to inform targeted development of novel therapeutics for prevention of acute reactions. In this study, we take an in vivo approach (Fig. 1) to comprehensively profile changes in the transcriptome that occur during the course of acute peanut allergic reactions. We apply RNA-sequencing (RNA-seq) to serial peripheral blood samples from a cohort of peanut allergic children before, during, and after randomized, double-blind, placebo-controlled oral peanut challenges. Using these data, we integrate results from linear mixed-effects (lme) models, leukocyte deconvolution, weighted gene coexpression network analysis (WGCNA), and probabilistic causal gene network modeling to identify causal key driver genes, biological processes, and cell types involved in acute peanut allergic reactions. Support for our major findings is then gained from replication in an additional 21 peanut allergic subjects.

## Results

### Clinical characteristics of the peanut allergic cohort. Twenty-one children with suspected peanut allergy completed randomized, double-blind, placebo-controlled oral food challenges to peanut, performed according to a modified AAAAI/EAACI PRACTALL protocol[8, 9]. Under close medical supervision with blinding of subjects and staff, subjects ingested incremental

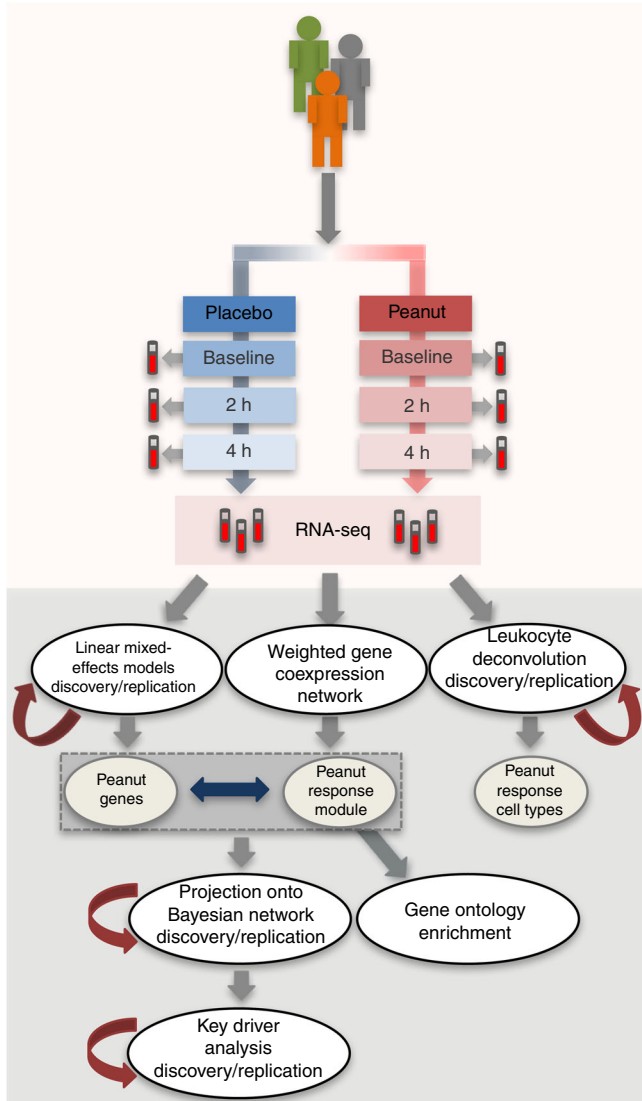

**Fig. 1** Overview of study design and analytical flow. Schematic outlining the clinical phenotyping, sample collections, RNA-sequencing, and analyses conducted. A discovery cohort consisting of 19 peanut allergic subjects, underwent physician-supervised, double-blind oral food challenges to both peanut and placebo, with the order of challenges randomized. For each subject, whole blood samples at baseline, 2 h, and 4 h were each collected during the peanut and placebo challenges. RNA-sequencing was performed on all samples and used for the following analyses outlined in gray: (1) linear mixed-effects model analyses to identify peanut genes (i.e., genes with expression changes in response to peanut, but not placebo, during acute peanut allergic reactions); (2) weighted gene coexpression network analysis (WGCNA) to identify modules of co-expressed genes as broader constructs of biologic function; and (3) leukocyte deconvolution analysis to identify cellular response during the acute peanut allergic reactions. Peanut genes identified by linear mixed-effects model analysis and WGCNA were projected onto a probabilistic causal gene network. Functional biologic processes implicated by WGCNA and peanut genes were investigated further using gene ontology enrichment analysis. Key driver analysis was implemented to identify key causal modulators of the peanut response module enriched for peanut genes. A replication cohort of 21 peanut allergic subjects analogously profiled was used to replicate major findings from this study (steps for which replication analyses were conducted are indicated with maroon arrows)

**Table 1 Characteristics of the peanut allergic discovery cohort (n = 19) and replication cohort (n = 21)**

| | Discovery Cohort (n = 19) | Replication Cohort (n = 21) |
|---|---|---|
| Sex: Female | 7 (36.8%) | 7 (33.3%) |
| Age: years | 12.0 (4.0) | 11.0 (5.0) |
| Parental allergy | 16 (84.2%) | 21 (100.0%) |
| Peanut sIgE: $kU_a/L$ | 68.0 (82.8) | 87.4 (140.5) |
| Peanut skin prick test: mm | 12.0 (6.0) | 13.0 (11.0) |
| Cumulative dose at first objective symptom: g peanut protein | 0.014 (0.44) | 0.044 (0.14) |
| Cumulative successfully consumed dose: g peanut protein | 0.144 (0.48) | 0.144 (0.40) |
| Symptoms experienced during peanut challenge | | |
| Distress | 13 (68.4%) | 14 (66.7%) |
| Throat tightness | 9 (47.4%) | 6 (28.6%) |
| Rhinorrhea | 3 (15.8%) | 2 (9.5%) |
| Rash | 1 (5.3%) | 4 (19.0%) |
| Rubbing of eyes, nose, or scratching | 2 (10.5%) | 3 (14.3%) |
| Urticaria: 1–2 lesions | 3 (15.8%) | 6 (28.6%) |
| Urticaria: >3 lesions | 2 (10.5%) | 5 (23.8%) |
| Angioedema | 4 (21.1%) | 9 (42.9%) |
| Vomit: single | 2 (10.5%) | 2 (9.5%) |
| Vomit: multiple times | 1 (5.3%) | 0 (0.0%) |
| Abdominal pain: severe | 2 (10.5%) | 5 (23.8%) |
| Diarrhea | 0 (0.0%) | 1 (4.8%) |
| Wheeze | 0 (0%) | 1 (4.8%) |
| Stridor | 1 (5.3%) | 0 (0.0%) |
| Hypotension | 0 (0%) | 0 (0.0%) |

Number (percent) or Median (IQR) are shown

amounts of peanut at 20 min intervals until an allergic reaction occurred or until the cumulative dose of 1.044 grams protein was ingested. In a similar fashion and on a different day, the same subjects ingested incremental doses of placebo oat powder. The order of peanut and placebo challenges was randomized, and all subjects completed both peanut and placebo challenges. Peripheral blood samples were drawn before, during, and at the end of each challenge.

Nineteen of the 21 children reacted to peanut, and none of the subjects reported symptoms during the placebo challenge; clinical characteristics are shown in Table 1. The two children who met inclusion criteria and completed the challenges, but had no reaction to peanut, were considered not peanut allergic and were excluded from further study. Among the 19 peanut allergic children, peanut sIgE ranged from 1.21 to >100 $kU_A/L$, and peanut skin prick test wheal ranged from 4.5 to 39.5 mm.

All children reacted within the first 2 h of the peanut challenge. Five children with peanut allergy reacted to the first dose of peanut, with symptoms of an allergic reaction following just 1 mg of peanut protein. The median cumulative dose at first objective symptom among the peanut allergic children was the third dose (10 mg dose for a cumulative intake of 14 mg peanut protein). The most common symptom elicited by peanut was distress, followed by throat tightness, urticaria, angioedema, vomiting, and rhinorrhea. All children received antihistamines and 58% were administered epinephrine.

**Identifying genes associated with peanut allergic reactions.** The primary aims of this study were to characterize gene expression signatures, functional processes, and causal key drivers of acute

peanut allergic reactions. To do this, during each peanut and placebo challenge, we collected peripheral blood from each subject at zero hours (baseline, pre-challenge), 2 h (during challenge) and 4 h (end of challenge) from challenge start, totaling 6 samples per subject for RNA-seq (Fig. 1). This design enabled us to execute analyses such that each subject served as their own control over time and exposure (peanut vs. placebo). Six samples were collected from each of the 19 subjects, except for two participants for whom samples were obtained during peanut challenge only, resulting in a total of 108 samples collected.

Following data processing, normalization, and quality control, one outlier sample was removed (Supplementary Figure 1), resulting in RNA-seq data (n = 17,337 genes) for 107 blood samples (placebo n = 51; peanut n = 56) across the 19 peanut allergic subjects. To identify genes associated with acute peanut allergic reactions, we used lme modeling[10], which allowed us to collectively and statistically assess, across all individuals in this peanut allergic cohort, within-individual changes in gene expression across all time points in response to peanut and not placebo (see Methods). The top 30 genes identified by this analysis (Bonferroni-corrected P < 0.01) are shown in Table 2; functional information for each gene is summarized in Supplementary Table 1. Because effect size estimates from lme models can be less intuitive, changes in gene expression between the means at baseline and four-hour time points for the peanut challenge are displayed in Table 2 as a more intuitive measure of effect size. As Bonferroni correction is often considered overly conservative, an extended set of genes significant at P < 0.005 was used for many of the downstream analyses that follow (Supplementary Data 1). For ease, we refer to this extended set of peanut allergen response genes henceforth as "peanut genes."

Gene expression levels for six selected peanut genes are plotted in Fig. 2 as examples of patterns observed for genes identified by lme modeling; these genes are also the six key drivers identified by causal network analyses conducted below. For context, plots for the top 30 peanut genes are provided in Supplementary Fig. 2. Consistent with the activation of pro-inflammatory processes, the majority of peanut genes were upregulated with peanut exposure, with 65% (1411/2168) exhibiting increased gene expression at four hours after initial exposure to peanut. Across all peanut genes, we observed a greater magnitude of gene expression change between baseline and four hours (mean differences of 0.24 $log_2$-counts per million (c.p.m.)) compared to baseline and two hours (mean difference 0.04 $log_2$-c.p.m.).

**Leukocyte compositional changes in peanut allergic reactions.** Given that the allergic response involves the activation, differentiation, and recruitment of various immune cell types, we tested for changes in the distribution of leukocyte cell fractions during acute peanut allergic reactions. Employing a leukocyte cell-type deconvolution algorithm[11], we inferred the proportions of 19 leukocyte populations from RNA-seq expression profiles of each sample and time point (Fig. 3a). Neutrophils, naive $CD4^+$ T cells, and monocytes were the predominant cell types observed, collectively representing 70% of cells on average across samples. In contrast, the remaining cell types each had individual mean fractions <10%.

Analogous to our gene expression analysis, we used lme models to identify leukocyte cell fractions with significant changes over time in response to peanut but not placebo. We observed significant differences in three specific cell types after correction for multiple testing (Fig. 3b): M0 macrophages (FDR = 0.016), neutrophils (FDR = 0.016), and naive $CD4^+$ T cells (FDR = 0.018). Whereas the fraction of both macrophages and neutrophils continuously increased over time during the peanut challenge,

**Table 2 Top 30 peanut allergen response genes (peanut genes) identified by linear mixed-effects models (Bonferroni-corrected $P < 0.01$) in the discovery cohort and their corresponding results in the replication cohort**

| Gene | Discovery Cohort ($n = 19$) | | | Replication Cohort ($n = 21$) | |
|---|---|---|---|---|---|
| | $P$-value | Bonferroni $P$-value | Peanut T4 vs. T0 $\Delta$ log$_2$-cpm[a] | $P$-value | Peanut T4 vs. T0 $\Delta$ log$_2$-cpm[a] |
| **PLP2** | $1.24 \times 10^{-08}$ | $2.00 \times 10^{-04}$ | 0.55 | $3.81 \times 10^{-03}$ | 0.38 |
| **LY9** | $1.43 \times 10^{-08}$ | $2.00 \times 10^{-04}$ | −0.48 | $4.34 \times 10^{-02}$ | −0.22 |
| **APLP2** | $1.57 \times 10^{-08}$ | $2.00 \times 10^{-04}$ | 0.62 | $1.68 \times 10^{-04}$ | 0.40 |
| **AGPAT9** | $3.21 \times 10^{-08}$ | $5.004 \times 10^{-04}$ | 1.18 | $9.95 \times 10^{-06}$ | 0.78 |
| **NOV** | $3.56 \times 10^{-08}$ | $6.00 \times 10^{-04}$ | 0.97 | $9.86 \times 10^{-03}$ | 0.45 |
| **DGKG** | $5.94 \times 10^{-08}$ | 0.001 | 0.73 | $8.85 \times 10^{-04}$ | 0.48 |
| **BST1** | $9.64 \times 10^{-08}$ | 0.0016 | 1.11 | $6.86 \times 10^{-04}$ | 0.68 |
| **CRK** | $1.05 \times 10^{-07}$ | 0.0018 | 0.54 | $4.01 \times 10^{-04}$ | 0.31 |
| CD22 | $1.09 \times 10^{-07}$ | 0.0018 | −0.55 | $3.89 \times 10^{-01}$ | −0.16 |
| **MOSC1** | $1.75 \times 10^{-07}$ | 0.003 | 1.37 | $7.29 \times 10^{-05}$ | 0.92 |
| PLXDC1 | $1.76 \times 10^{-07}$ | 0.003 | −0.31 | $2.85 \times 10^{-01}$ | −0.15 |
| **PADI4** | $1.93 \times 10^{-07}$ | 0.0033 | 1.51 | $4.21 \times 10^{-04}$ | 0.86 |
| **FCAR** | $1.95 \times 10^{-07}$ | 0.0033 | 1.49 | $5.68 \times 10^{-04}$ | 1.05 |
| **DDX55** | $2.22 \times 10^{-07}$ | 0.0038 | −0.22 | $2.83 \times 10^{-02}$ | −0.17 |
| **PLCG2** | $2.66 \times 10^{-07}$ | 0.0046 | 0.55 | $1.99 \times 10^{-03}$ | 0.28 |
| **SULF2** | $2.75 \times 10^{-07}$ | 0.0047 | 0.73 | $1.95 \times 10^{-06}$ | 0.61 |
| **LTB4R** | $2.83 \times 10^{-07}$ | 0.0049 | 1.21 | $2.03 \times 10^{-03}$ | 0.51 |
| **RBP7** | $2.86 \times 10^{-07}$ | 0.0049 | 0.97 | $1.81 \times 10^{-03}$ | 0.68 |
| **OSCAR** | $2.92 \times 10^{-07}$ | 0.005 | 0.77 | $1.69 \times 10^{-03}$ | 0.50 |
| **MMP9** | $3.29 \times 10^{-07}$ | 0.0057 | 2.24 | $2.79 \times 10^{-04}$ | 1.36 |
| **JDP2** | $3.42 \times 10^{-07}$ | 0.0059 | 1.26 | $5.88 \times 10^{-04}$ | 0.75 |
| **ACSS2** | $3.49 \times 10^{-07}$ | 0.006 | 0.52 | $7.17 \times 10^{-04}$ | 0.26 |
| **CCDC164** | $3.50 \times 10^{-07}$ | 0.006 | 1.67 | $3.07 \times 10^{-04}$ | 1.22 |
| **EPHB4** | $3.70 \times 10^{-07}$ | 0.0064 | 1.31 | $1.86 \times 10^{-05}$ | 0.81 |
| **C5orf39** | $3.89 \times 10^{-07}$ | 0.0067 | −0.21 | $1.11 \times 10^{-02}$ | −0.18 |
| **SLC26A8** | $4.14 \times 10^{-07}$ | 0.0071 | 1.27 | $2.3 \times 10^{-05}$ | 0.92 |
| **DIRC2** | $4.18 \times 10^{-07}$ | 0.0072 | 0.76 | $3.73 \times 10^{-04}$ | 0.53 |
| **BMX** | $4.58 \times 10^{-07}$ | 0.0079 | 2.05 | $6.58 \times 10^{-05}$ | 1.28 |
| **LILRA3** | $4.66 \times 10^{-07}$ | 0.008 | 1.26 | $1.69 \times 10^{-03}$ | 0.63 |
| **FCRL3** | $5.27 \times 10^{-07}$ | 0.009 | −0.43 | $3.67 \times 10^{-03}$ | −0.45 |

[a]Because effect size estimates from lme models are less intuitive, changes in gene expression between the means at baseline and four-hour time points for the peanut challenge are displayed as a more intuitive estimate of effect size; Bolded gene names are those for which significant $P$-values were observed in both the Discovery and Replication cohorts

but not placebo, there was a reduction in naive CD4$^+$ T cells at four hours compared to baseline (Fig. 3b). Raw and corrected $P$-values from this analysis are provided in Supplementary Table 2.

To treat severe reactions, some subjects in our study ($n = 11$) were given epinephrine during peanut challenge. Because epinephrine can impact cell numbers in peripheral blood[12, 13], we stratified our data to assess whether the same trends were observed in subjects who did and did not receive epinephrine (Supplementary Fig. 3). Consistent with the combined results, we observed increased fractions of macrophages and neutrophils, and a decreased fraction of naive CD4$^+$ T cells, after peanut challenge in both the epinephrine treated and untreated strata. However, particularly for neutrophils and CD4$^+$ T cells, the effects were more pronounced in subjects who received epinephrine, which could be attributable to the fact that these individuals had more severe allergic reactions. To directly test whether this difference was due to reaction severity vs. epinephrine treatment, an experimental design barring epinephrine treatment in subjects with severe reactions would be needed; however, this is not ethically possible in human subjects given the risk of anaphylaxis and death with untreated severe allergic reactions.

**Replication of gene expression and leukocyte signatures**. To assess the robustness of the changes in gene expression and leukocyte fractions observed, we sought to replicate our findings in an independent replication cohort of 21 peanut allergic children, with clinical characteristics similar to those of the discovery cohort (Table 1). Each of the 21 individuals underwent analogous double-

blind, placebo-controlled peanut challenges using the same sample collection protocol as the discovery cohort. We used RNA-seq data generated from whole blood collected at baseline, two hours, and four hours of the peanut challenge to test for gene expression changes that occurred in response to peanut. Of the top 30 genes identified in the discovery cohort, 28 exhibited statistically significant ($P < 0.05$) changes in expression during peanut challenge in the replication cohort (Table 2), representing a significant enrichment (OR = 27.9, Fisher's exact test $P = 9.5 \times 10^{-12}$) over that expected by chance. Boxplots depicting expression data in this replication cohort for each of the top 30 genes presented in Table 2 are shown in Supplementary Fig. 4, showing concordant directions of gene expression change for all genes during peanut challenge relative to the discovery cohort.

We next used transcriptome profiles from the replication cohort to perform peripheral blood leukocyte deconvolution (Supplementary Fig. 5a) and lme modeling to identify changes in the proportions of cell subsets during peanut challenge. The same three cell types identified in the discovery cohort were again significant in the replication cohort (Supplementary Fig. 5b; M0 macrophages (FDR = 0.0053); neutrophils (FDR = 0.0013); naive CD4$^+$ T cells (FDR = 0.0038)).

**Peanut genes enriched in an acute-phase response module**. To investigate the biological context and relationships of peanut genes identified by our analyses, we constructed transcriptome-wide gene networks in the discovery cohort using WGCNA[14]. WGCNA identifies modules of co-expressed genes (i.e., groups of

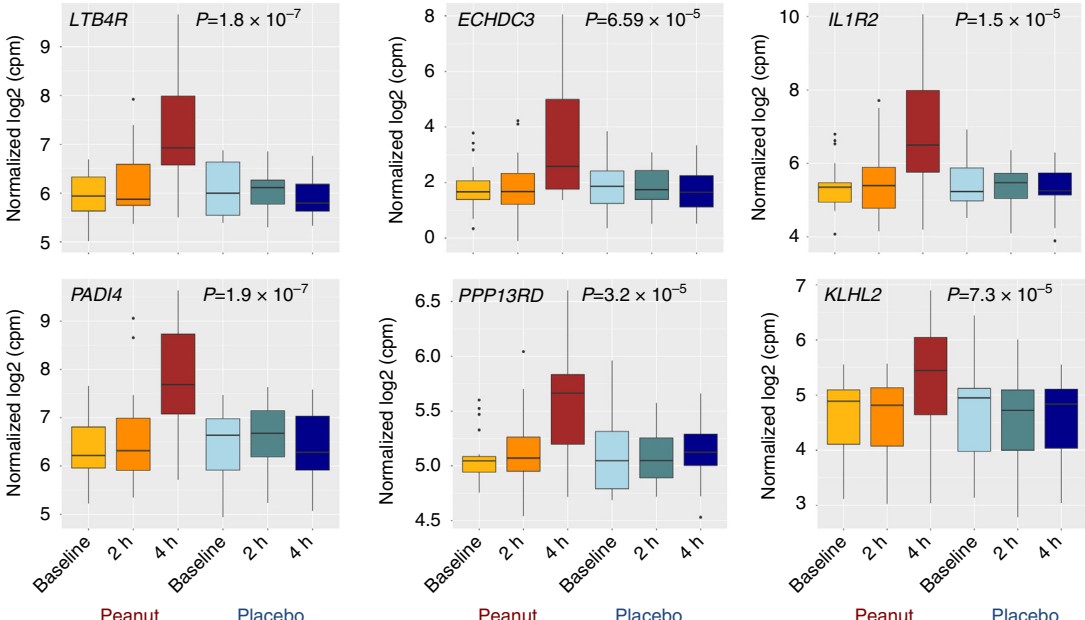

**Fig. 2** Peanut genes exhibit significant changes in gene expression following challenge to peanut but not placebo. Boxplots displaying $\log_2$-cpm expression values for six selected peanut genes in the 19 peanut allergic discovery cohort subjects at three time points (baseline, during challenge (2 h), and end of challenge (4 h)) for both peanut and placebo challenges. The majority of the peanut genes we identified, including the six selected genes shown, exhibited increased expression with peanut challenge. To identify genes with expression changes that occurred specifically during peanut but not placebo challenge, we used linear mixed-effects models to test for a significant contribution of the interaction between time and peanut exposure on gene expression. *P*-values for the likelihood ratio test between the test model and null lme models are shown for each gene

genes with similar expression profiles and interconnectivity across experimental samples), that can serve as broader constructs of gene expression beyond the single gene level[15]. Using WGCNA, we identified 13 coexpression modules (Supplementary Table 3). To identify modules associated with acute peanut allergic reaction, we tested for enrichment of peanut genes in each module. Only the blue coexpression module was significantly enriched for peanut genes. 1223 (51%) of the 2381 genes in that module were peanut genes, equivalent to a 4.1-fold enrichment over that expected by chance (Fisher's exact test $P = 1.2 \times 10^{-304}$) (Fig. 4a; Supplementary Table 3). We will henceforth refer to the blue module as the "peanut response module," given it was the only module enriched for peanut genes.

To gain insight into the collective putative function of genes within the peanut response module, we next performed gene ontology (GO) analysis[16]. This revealed significant enrichments of the peanut response module for inflammatory processes, including acute-phase response (fold enrichment = 3.5; FDR = $6.5 \times 10^{-3}$), acute inflammatory response (fold enrichment = 2.8, FDR = $2.9 \times 10^{-3}$), positive regulation of I-kappa-B kinase/NF-kappa-B signaling (fold enrichment = 1.9; FDR = $1.8 \times 10^{-3}$), and lymphocyte activation (fold enrichment = 1.7; FDR = $3.0 \times 10^{-3}$). The GO biological process terms associated with the peanut response module at FDR < 0.01, sorted by fold enrichment, are shown in Fig. 4b. A complete list of these GO terms and associated genes are provided in Supplementary Data 2. Although no other coexpression module identified by WGCNA was enriched for peanut genes after correction, we show the top biological processes associated with these other modules for comparison (Fig. 4a). To further enhance our understanding of the peanut response module, we examined GO biologic process terms for the upregulated and downregulated peanut genes in this module separately (Fig. 4c, d). Whereas the upregulated genes are involved in inflammation, the downregulated genes regulate

macromolecule metabolism; *P*-values and enrichment statistics from this analysis are provided in Supplementary Data 3.

**Identifying causal key drivers of the peanut response module.** Although integration of our peanut gene analysis with WGCNA provided strong evidence for a link between acute peanut allergic reactions and the peanut response module, this type of analysis is associative and thus cannot on its own reveal causal relationships among genes in the implicated module. Elucidating the connectivity structure within these modules can lead to the identification of key driver genes that are predicted to modulate the regulatory state of the module, and be of high interest to prioritize as causal to acute peanut allergic reactions. We have previously integrated genetic, transcriptome and other high-throughput data into probabilistic causal gene networks as one approach to uncover key driver genes in disease associated networks/modules[17, 18]. This type of data-driven, directed approach provides systems-wide context for understanding known and novel regulators and processes for a disease. We sought to apply such approaches here to identify key driver genes of the peanut response module.

Given the lack of large-scale gene expression data sets generated from individuals with peanut or other food allergies, which would be needed to construct accurate probabilistic causal gene networks[19], we explored the availability of alternative data sets that could be used to model gene networks associated with inflammation (consistent with processes identified by WGCNA analyses above). We constructed a directed probabilistic causal gene network using peripheral blood gene expression data and expression quantitative trait loci (eQTL) from 526 patients with inflammatory bowel disease (IBD)[20]. Although IBD is distinct from food allergy, both conditions are due to maladaptive immune reactions to antigens presented to the gastrointestinal tract in genetically susceptible individuals, and there is an

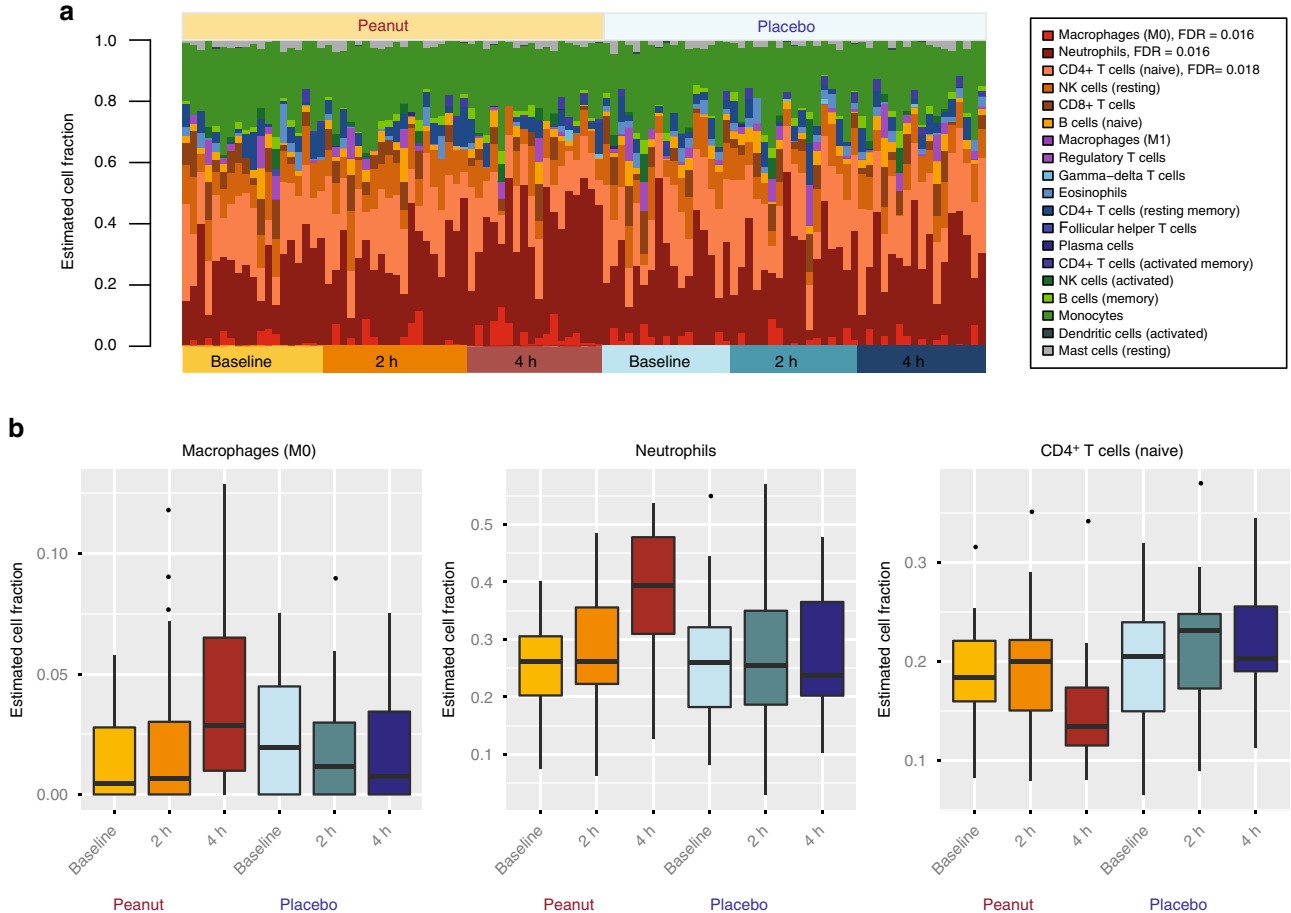

**Fig. 3** Peanut challenge induces compositional changes in leukocyte populations involved in the acute-phase response. **a** Fractions of leukocyte subpopulations estimated from transcriptome-wide RNA-seq gene expression signatures from the discovery cohort ($n = 19$), partitioned by challenge and time point. The order in which individuals are plotted is consistent across time/challenge groups. Changes in cell-type composition associated with peanut challenge were assessed using linear mixed-effect models. Cell types included in the analysis are indicated in the legend, listed in ranked order according to significance. FDR values are provided for the three cell types that exhibited significant changes in response to peanut but not placebo (macrophages (M0); neutrophils; naive CD4+ T cells). Data for these significant peanut response cell types are plotted in **b**, again partitioned by challenge and time point

association between IBD and atopic diseases, including food allergy[21, 22]. We reasoned that there is some overlap between the two conditions with regard to chronic immune dysregulation and pro-inflammatory processes such that the topology and major edges of the underlying disease network architecture in IBD could inform gene coregulation relevant to the inflammatory response to peanut.

To narrow in on the component of the constructed probabilistic causal gene network most relevant to acute peanut allergic reactions, we projected peanut genes belonging to the WGCNA peanut response module onto this network. The projection consisted of overlapping genes from the intersection of these genes and all genes in the network, identifying all genes in the network that are within a path length of one node in this overlap, and then identifying the largest connected graph (subnetwork) from this set of nodes and all associated edges. The subnetwork identified from this projection was composed of 1029 genes, including 500 genes belonging to both the peanut response module and peanut gene set, representing a 4.7-fold enrichment over that expected by chance (Fisher's exact test $P = 3.81 \times 10^{-272}$). This indicated that the peanut response module constructed from our study was highly conserved in the probabilistic causal network constructed from the IBD cohort.

To predict genes that modulate the regulatory state of the peanut response module, we employed key driver analysis (KDA), an algorithm that mathematically identifies causal modulators of the regulatory state of functionally relevant gene groups[17]. Key drivers have been found to be reproducible and successfully validated in different contexts[23]. KDA identified 26 key drivers (FDR < 0.05) of the peanut response module in the IBD network, and these key drivers were significantly enriched for genes in the identified subnetwork (OR = 82.6, Fisher's exact test $P = 9.65 \times 10^{-20}$) (Supplementary Table 4). The relationship between key drivers with the peanut genes, and genes belonging to the peanut response module are depicted in Fig. 5. This hierarchical structure of the network, reflects an ordering of the key driver genes in which key drivers at the highest level have the most upstream effects on modulating the state of genes at the lowest level, which specifically comprise peanut genes and genes belonging to the peanut response module. As the distance from the bottom increases, so too does the enrichment at each level for being a key driver. The higher up in the response cascade, the higher the odds ratio (OR 274.1, Fisher's exact test $P = 6.0 \times 10^{-16}$ for the top level). Resolving the structure of these networks allows for the identification of those driver genes that are causally associated with the expression of peanut genes at the most upstream points in the response cascade.

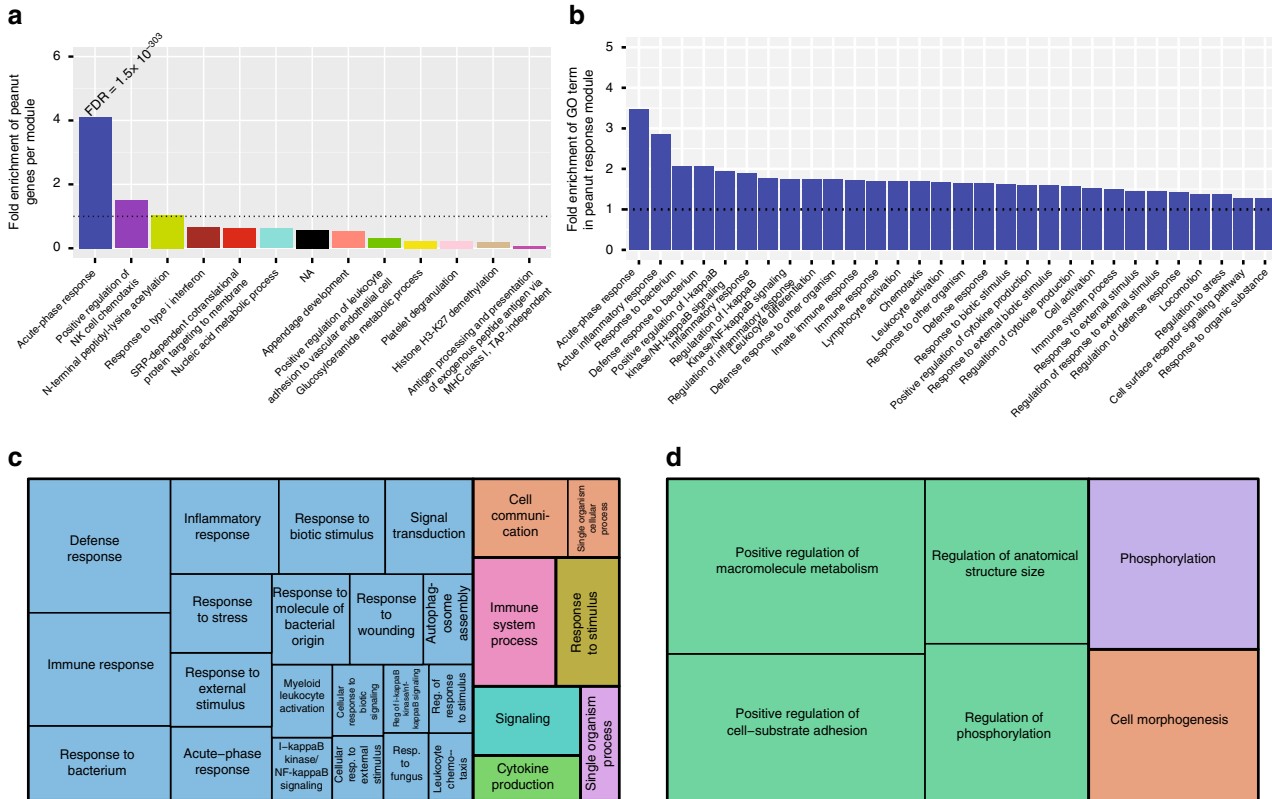

**Fig. 4** Identification and characteristics of the peanut response coexpression module. **a** Fold enrichment of peanut genes ($P < 0.005$) in the 13 coexpression modules identified by WGCNA, ranked from highest enrichment to lowest. The FDR (Fisher's exact test) for the most highly enriched module is shown (peanut response module). Gene ontology (GO) enrichment analysis[17] was conducted on each module; the top 30 most significant GO terms for each module were ranked by fold enrichment, and the highest-fold term for each module is shown on the *x* axis. No significant terms were found for genes in the black module. **b** The top thirty GO terms associated with the peanut response module, ranked by fold enrichment. Horizontal dotted lines in both **a** and **b** correspond to a fold enrichment = 1. **c** GO terms associated with upregulated genes (FDR ≤ 0.05) in the peanut response module, with size of box inversely corresponding to FDR. **d** GO terms associated with downregulated genes (FDR ≤ 0.05) in the peanut response module, with size of box inversely corresponding to FDR

To ensure that the IBD network was capturing biology relevant to peanut allergy, we constructed a validation causal gene network using the peanut allergic discovery and replication cohorts (Table 1). As context, we used the IBD network as our discovery causal network because it could be built on a much larger data set ($n = 526$) with genetic prior information (expression quantitative trait loci (eQTL)), thus enabling construction of a robust and informative causal network[19, 24, 25]. The peanut causal network we built is not as robust as the IBD network given the limited sample size for its construction and lack of eQTLs available as input, and was therefore used to validate findings from the IBD network. Indeed, when we performed the same subnetwork analysis as above but in the peanut network, the overlap between the IBD and peanut subnetworks was 3-fold enriched over that expected by chance (Fisher's exact test $P = 3.3 \times 10^{-224}$). Further, KDA of the peanut allergy-specific causal network identified key drivers significantly overlapping those from the IBD network (OR 16.8, Fisher's exact test $P = 2.4 \times 10^{-7}$), lending confidence to our causal network findings.

Six key driver genes (*LTB4R*, *PADI4*, *IL1R2*, *PPP1R3D*, *KLHL2*, and *ECHDC3*) in particular were highlighted by the combined interpretation of the lme models, WGCNA, and KDA, as these were the only genes that met all three criteria of being: (1) peanut genes with a significant change in expression with peanut but not placebo exposure; (2) members of the WGCNA peanut response module; and (3) top level key driver genes in the hierarchical visualization of the network, predicted to causally modulate the

state of peanut genes and peanut response module members at the most proximal level (Fig. 5). To illustrate the relationship between these six key drivers of highest interest, Fig. 6 shows their relationship within the probabilistic causal gene network, as well as their cellular context based on prior knowledge. Two of these top layer key drivers, *LTB4R* and *PADI4*, were peanut genes by the strictest significance criteria (Bonferroni-corrected $P < 0.01$) (Table 2; Fig. 2). In addition, *LTB4R* and *PADI4* have previously demonstrated roles in inflammatory and immune-related diseases[26, 27], as does *IL1R2*, another top layer key driver[28, 29].

## Discussion

In this study, we collected and analyzed serial peripheral blood transcriptomic profiles from a cohort of peanut allergic children undergoing double-blind, placebo-controlled oral food challenges to identify six causal key driver genes (*LTB4R*, *PADI4*, *IL1R2*, *PPP1R3D*, *KLHL2*, and *ECHDC3*) of high interest in acute peanut allergic reactions. Analysis of a second peanut allergic cohort of 21 children replicated our major findings. The results of this study provide a data-driven, circumscribed set of high-yield gene targets for directed mechanistic studies of peanut allergy, the most common cause of food-related anaphylaxis and death in the United States.

Currently, allergen avoidance and prompt care of allergic reactions remain the standard treatment strategy for food allergic individuals. Although immunotherapy for desensitization and/or

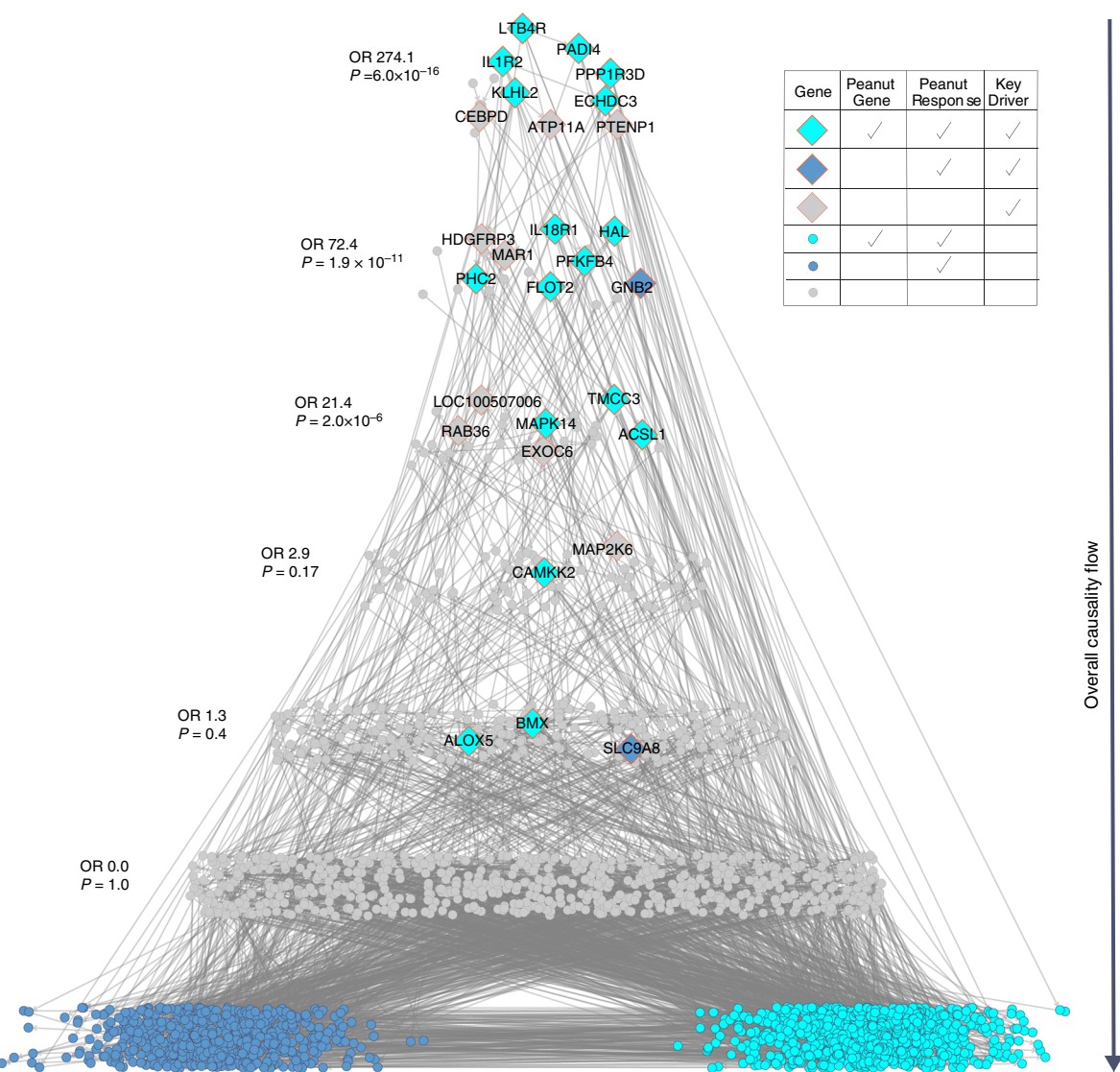

**Fig. 5** Probabilistic causal gene network projection and key driver analysis identifies causal regulators of the peanut response module. Eiffel tower plot depicting the probabilistic causal gene network for peanut response genes and its key drivers. The overall direction of causality flow is indicated by the arrow to the right of the tower; genes at the highest levels have the greatest causal impact on the expression of genes in this network. Significant key drivers (FDR ≤ 0.05) in the network are depicted as filled diamonds, with gene name labels. All other genes in the network are shown as filled circles. As per the legend, genes and key drivers are colored according to their designation as peanut response module gene (dark blue), both a peanut response module and peanut gene (turquoise), or neither (gray). Odds ratios and P-values for the enrichment of genes at each path level (corresponding to path length) for key drivers are shown to the left

sustained unresponsiveness have demonstrated progress, immunotherapy is not effective for all individuals[30–32], carries significant adverse side effects, may not have long-term efficacy, and is not ready for generalized clinical implementation[33]. In addition, the fact that unexpected acute reactions are responsible for the majority of mortalities associated with food allergy, stresses the need for better understanding the molecular processes mediating such reactions.

A major strength of this study is that the design (Fig. 1) allowed for rigorous characterization of changes in gene expression signatures that occurred during allergic reactions to peanut by profiling the transcriptomes of peanut allergic subjects at multiple time points throughout the course of randomized, double-blind, placebo-controlled challenges. Although such challenges are considered the gold standard for characterizing food allergy, they are not routinely performed because they are resource-intensive. Rather than analyzing differences between

allergic and non-allergic individuals (i.e., case-control design used in most studies of food allergy to date), our design allowed us to leverage data collected at multiple time points and between in vivo exposures (peanut and placebo) within the same individual, increasing precision and power by minimizing bias due to inter-individual variability, and optimizing relevance of findings to actual human systems. Further, we were able to replicate our major findings in a second cohort of 21 peanut allergic children.

We identified peanut genes (Table 2, Supplementary Table 2) that exhibited significant changes in expression in response to challenge with peanut but not placebo. Asserting the robustness of our results, an overwhelming majority (93%) of the top peanut genes identified in our discovery cohort were replicated in a second independent cohort. We interpret the observed expression signatures dominated by upregulation to be consistent with activation of the acute-phase response to peanut allergen. Indeed, the most significant peanut genes, including key drivers, are

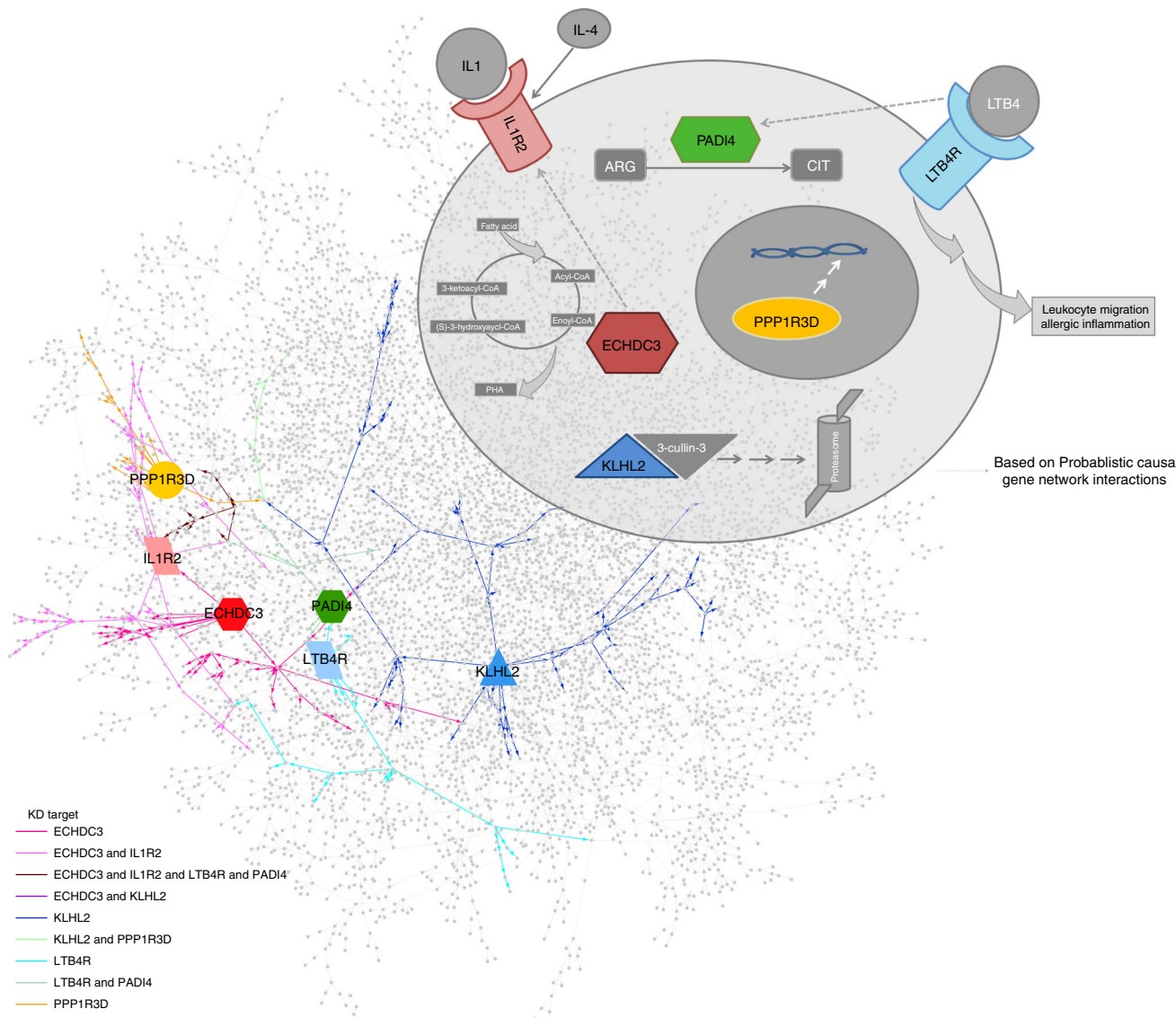

**Fig. 6** Key drivers interact within the probabilistic causal gene network and cellular environment. A cartoon cell schematic of the key drivers identified as primary causal regulators of the acute-phase response module and peanut response genes is shown in the upper right, demonstrating their locations of activity in the cellular context based on prior knowledge. Activation of LTB4R by LTB4 binding leads to macrophage, T cell, and neutrophil chemotaxis. PADI4 converts arginine (ARG) to citrulline (CIT) residues and plays a role in granulocyte and macrophage development. Induced and released by IL4, IL1R2 is a decoy receptor that inhibits IL1 activity. ECHDC3 is an enzyme involved in fatty acid biosynthesis. PP1R3D regulates protein serine/threonine phosphatase activity, and KLHL2 is involved in proteasomal degradation and reorganization of actin cytoskeleton for cell projection by oligodendrocyte precursors. At the bottom left, the constructed probabilistic causal gene network is displayed with key drivers indicated by enlarged, labeled nodes, with their shape and color corresponding to the cell schematic. Edges are colored based on the interaction of each key driver with downstream genes at a path length of seven, displaying the singular and combinatorial downstream effects that each key driver can have on this network. Interactions between the key drivers inferred from the probabilistic causal gene network are indicated in the cell schematic by the dashed-line arrows

expressed by inflammatory cells with roles in immune activation, immune effector functions, chemokine and cytokine signaling, and cell migration and proliferation[26–29, 34–37]. These expression trends and functional categories are similar to those recently shown in a gene expression study of six subjects admitted to the emergency room with anaphylaxis (including two individuals for whom peanut exposure was the cause)[38]. Investigation in cohorts with other food allergies will be required to determine the extent to which expression of peanut genes are specific to the allergic response to peanut.

As genes do not function in isolation, we constructed weighted coexpression and probabilistic causal gene networks to model molecular interactions and causal gene relationships, and applied mathematical approaches (KDA) to identify and prioritize key drivers of acute peanut allergic reactions. Indeed, effects of peanut exposure on gene expression extended beyond single genes, influencing networks of co-regulated loci involved in acute-phase response and inflammatory processes. Specifically, peanut genes were significantly over-represented in a single coexpression module identified using WGCNA (Fig. 4). This peanut response module of 2381 genes was highly enriched for pro-inflammatory processes, including regulation of the Rel/nuclear factor-κB (NF-κB) transcription factor family. NF-κB signaling has been investigated extensively in asthma and allergy, and shown to regulate key cytokines, chemokines, and cell adhesion molecules in immune cells infiltrating sites of inflammation[39, 40]. A recent

study employing WGCNA on transcriptomes of dust mite-stimulated CD4[+] T cells from asthmatic subjects found *NFKBIA* (a member of the NF-κB family) to be one of nine hub genes within the primary coexpression module identified[41]. This gene and several others related to NF-κB regulation have also been associated with genetic risk of asthma and related phenotypes[42]. In contrast to other allergic diseases, NF-κB signaling in food allergy has been less studied; however, its activation was recently observed in a mouse model of intestinal anaphylaxis following ovalbumin challenge[43]. Expression profiling in neonatal CD4[+] T cells of food allergic individuals also revealed the dysregulation of several genes involved in NF-κB signaling[4].

Our probabilistic causal network and KDA identified and prioritized six key drivers that most strongly and causally modulated the peanut response module in the network (Fig. 5). While little is known about *KLHL2*, *ECHDC3*, and *PPP1R3D* in the immune response, *LTB4R*, *PADI4*, and *IL1R2* have established roles in inflammation. *LTB4R* encodes a receptor for leukotriene B4 (LTB4) (Fig. 6), a lipid pro-inflammatory mediator involved in the recruitment of leukocytes to sites of inflammation, including neutrophils and macrophages[26, 37, 44]. Linked to biological processes implicated by GO analyses, LTB4R and its ligand influence the activation of NF-κB signaling[44, 45]. In addition, LTB4 and LTB4R have previously been implicated in the pathogenesis of atopic diseases, including asthma, atopic dermatitis, and allergic rhinitis[26, 46]. *PADI4*, although not explicitly implicated in processes associated with the allergic response, is a primary genetic risk factor for rheumatoid arthritis (RA)[27], a disease characterized by systemic inflammation. The expression of *PADI4* by various inflammatory cells, namely neutrophils[27, 47, 48], is elevated in inflamed synovial RA tissue, and associated with increased levels of RA-associated autoantibodies in humans and animal models[49–51]. *IL1R2* is expressed by various immune cells, and generally considered to be a negative regulator of pro-inflammatory IL-1 cytokine signaling, with roles in the production of interleukins, as well the activation of NF-κB signaling[52]. Upregulation of *IL1R2* has been observed in peripheral blood mononuclear cells of adult food allergic individuals[5]. In addition, variants in *IL1R2* have previously been implicated in atopic disease[28, 42], as well as other inflammatory disorders[52]. Our probabilistic causal gene network analysis additionally provided support for a directed relationship between *ECHDC3* and *IL1R2* (Fig. 6).

Our study was intentionally designed to not focus on any one peripheral blood cell-type, as multiple cell types have been implicated in acute food allergic reactions, and discovery thus far has been biased toward cell types sufficiently abundant for isolation. Our leukocyte deconvolution approach was unbiased so as to inclusively capture cellular expression signatures across peripheral blood. Given limitations on the volume of blood that can be collected from children, especially when sampling serially, flow cytometry-based assays to target multiple cell populations at each time point was not feasible. Our study provides data on 19 inferred leukocyte cell fractions (Fig. 3), highlighting three cell subsets that could be directly profiled in future studies. Importantly, each of the significant cell subset changes were replicated in our second peanut allergic cohort, bolstering the likelihood that these changes represent genuine signatures associated with acute allergic response to peanut. Among these, we observed that naive CD4[+] T cells decreased during peanut challenge. This is perhaps not surprising, given that food allergy is classically considered to be an IgE-mediated response orchestrated by Th2 T cells, which involves the activation of naive CD4[+] T cells and their transition to effector T cell types[3]. Decreases in naive CD4[+] T cells could also reflect their migration from the periphery to sites of inflammation. In contrast to the decrease of naive T cells, the proportions of neutrophils and M0 macrophages both increased in response to peanut, with neutrophils representing ~40% of cells on average in subjects four hours following exposure. Both neutrophils and macrophages hold primary roles in inflammation and allergic disease[53, 54]. For example, depletion of macrophages and neutrophils in several mouse models influences symptoms of peanut-induced anaphylaxis[55–57]. In humans, markers of neutrophil activation, such as plasma myeloperoxidase concentration, are significantly elevated in individuals experiencing acute anaphylaxis[58]. Interestingly, connecting cell-type analyses to results from KDA, it is notable that the expression of *LTB4R*, *PADI4*, and *IL1R2* are linked to the function of macrophages and neutrophils in the context of the inflammatory response[26, 49, 59–61]. For example, inhibition of *LTBR4* reduces macrophage chemotaxis and inflammation[59]. Likewise, the infiltration of neutrophils and recruitment of effector T cells to allergen-challenged sites is dependent on the expression of *LTB4/LTB4R* by neutrophils[26]. In contrast, the expression of *IL1R2* by neutrophils has been shown to mediate pro-inflammatory effects of IL-1[60]. These results support the targeting of these cell types for profiling of cell type-specific gene expression signatures in acute peanut allergy.

Because the key drivers, molecular processes, and cell types highlighted here were characterized in the context of changes occurring directly during allergic reactions, they have potential to inform the design and application of therapeutic strategies for mitigating symptoms of peanut allergic reactions. Notably, a subset of genes and biological processes highlighted by our integrative analyses already serve as therapeutic targets for other atopic and inflammatory diseases. For example, the key driver, *LTB4R*, is modulated by 5-lipoxygenase inhibitors, which are in clinical use for asthma; several LTB4R antagonists have also been developed[62]. A biological process identified by our integrated analysis, NF-κB signaling, is mediated by glucocorticoids[63], which are frequently administered for food allergic reactions. Also, allergen-specific subcutaneous immunotherapy for environmental allergies has been shown to suppress NF-κB signaling in neutrophils of allergic individuals[64]. Several other inhibitors and modulators of NF-κB signaling are in various stages of investigation and development[63, 65]. It would be interesting to investigate whether current treatment strategies can modulate peanut genes and biological networks identified here, and generalize to other food allergies as well.

In summary, we comprehensively profiled peripheral blood transcriptomic changes in peanut allergic individuals during acute allergic reactions. By adopting an integrative, data-driven approach, we have identified key driver genes, biological processes, and cell types associated with acute peanut allergic reactions. Furthermore, we were able to replicate our major findings in a second cohort. Our identification of six key driver genes includes three previously associated with other allergic and atopic diseases, as well as genes that remain less explored to date. This combination of findings serves as a useful sanity check of our results and indicates promise in targeting these genes and biological processes to further our mechanistic understanding of peanut allergy, with the potential to inform development of new treatment strategies.

## Methods

**Recruitment and sample collection**. Subjects age 4–25 years with physician-diagnosed peanut allergy or convincing history of peanut allergy with skin prick test positive to peanut (wheal diameter ≥3 mm greater than saline control) or detectable peanut specific IgE (ImmunoCAP >0.35 kU$_A$/L) were eligible to participate. Nineteen subjects were recruited from those undergoing screening oral food challenges to determine eligibility for the CoFAR6 study[9], a trial of epicutaneous immunotherapy for peanut allergy at five study centers across the United States including the Icahn School of Medicine at Mount Sinai, New York; University of North Carolina Medical Center, Chapel Hill, North Carolina; Johns Hopkins

University School of Medicine, Baltimore, Maryland; National Jewish Health, Denver, Colorado; and Arkansas Children's Hospital, Little Rock, Arkansas. These nineteen subjects were the first CoFAR6 subjects undergoing screening oral food challenges who consented to having additional blood drawn for transcriptomic study. An additional two subjects were recruited from patients undergoing oral food challenges as part of their clinical care of peanut allergy at the Icahn School of Medicine at Mount Sinai, New York, NY during the same time period. The institutional review boards of the participating institutions approved this study. Written informed consent was obtained from all participants.

All subjects underwent randomized, double-blind, placebo-controlled oral food challenges according to a modified AAAAI/EAACI PRACTALL protocol[8, 9]. Following randomization by coin flip and under close medical supervision with blinding of subjects and staff, each subject ingested incremental amounts of peanut at 20 min intervals until allergic reaction or until final cumulative dose of 1.044 grams protein ingested on one day, and ingestion of equivalent incremental doses of placebo oat powder in similar fashion on another day shortly before or after. The standardized doses ranged from 1 mg to 600 mg protein. The order of peanut and placebo challenges was randomized between subjects, with subjects completing both peanut and placebo challenges. Intravenous catheters allowed for 2.5 ml peripheral blood samples to be drawn right before challenge (baseline), during the challenge (2 h from challenge start), and at the end (4 h from challenge start) of the peanut and placebo challenges respectively into Tempus tubes. Fixed collection time points were chosen to ensure uniformity in collection strategy across peanut and placebo challenges performed by varying blinded study personnel at different sites. The rationale for sampling at 2 h is that experience at our study sites supports that most children who react during a peanut challenge exhibit symptoms within the first 2 h. We additionally sampled at 4 h because we hypothesized that adaptive immune responses could take longer to manifest as gene expression changes, and 4 h is a typical length of time for a food challenge visit to conclude.

Once a subject reacted with dose-limiting symptoms, the challenge was terminated with detailed documentation of the cumulative dose at first objective symptom, cumulative successfully consumed dose, and symptoms experienced by the subject. Medications including antihistamines, epinephrine, IV fluids, IV steroids, and beta-agonists were administered to the subject as needed according to the study protocol, and the subject was closely monitored until all symptoms resolved.

For the replication cohort, 30 additional children who also underwent double-blind, placebo-controlled peanut challenges as part of the CoFAR6 trial were eligible. Written informed consent was also obtained from these subjects with IRB approval from the same institutions as the discovery cohort. Serial whole blood collections were obtained following the same protocols used for the discovery cohort. Subjects who did not react to peanut ($n = 5$) or did not have samples collected during peanut challenge ($n = 4$) were then excluded, yielding a replication cohort of 21 peanut allergic subjects.

**Sample library preparation, sequencing, and data processing**. RNA was isolated from peripheral blood samples using Tempus Spin RNA Isolation kits (Applied Biosystems). For the discovery cohort, all available samples from peanut allergic children underwent RNA-seq library preparation and sequencing. For the replication cohort, only samples from the peanut challenge of peanut allergic children were processed and sequenced based on results from the discovery cohort demonstrating that the placebo-related samples exhibited unremarkable gene expression profiles (Supplementary Figure 2). Sequencing libraries for samples from the discovery and replication cohorts were processed and sequenced separately.

Following depletion of mitochondrial and cytoplasmic rRNA using Ribo-Zero Gold rRNA Removal (Illumina), mRNA libraries were sequenced on the Illumina HiSeq 2500 System with a per-sample target of 40–50 million 100 bp paired-end reads, according to the manufacturer's protocol. Read data were subjected to quality control with FastQC and RNA-SeQC. The data in fastq format were mapped to GRCh37 using the STAR v2.4.0g1 aligner (https://github.com/alexdobin/STAR), and gene-level counts were estimated using HTSeq (https://pypi.python.org/pypi/HTSeq).

Prior to statistical analyses, raw gene read counts were converted to cpm using the edgeR[66] package within R (www.R-project.org). Genes with systematically low expression levels were removed, keeping only genes with c.p.m. > 0.1 in at least 10% of the samples analyzed ($n = 17,337$). Library sizes and normalization factors were estimated, and counts were then normalized using edgeR and voom in the limma R package[67, 68], resulting in log₂-cpms and weights for each gene. Multidimensional scaling analysis was used to identify outlier RNA-seq samples, resulting in removal of one sample (Supplementary Fig. 1) from the discovery cohort. In addition, we used variancePartition[69], to assess the contribution of technical and biological factors to variation in gene expression across the samples. This revealed minimal influence of age and gender (Supplementary Fig. 6), and thus we did include these variables as covariates in the lme models next described.

**Identifying genes associated with acute peanut reactions**. For the 19 subjects in the discovery cohort who reacted to peanut during their double-blind, placebo-controlled food challenges, null and test lme models were constructed for each

gene, using the lme4 R package[10]. Each model included factor variables, "Time" (baseline, two hours, and four hours), "Challenge" (peanut or placebo), "Order" (which challenge came first), and "Weights" (voom output) as fixed effects, and per subject variation and collection date as random effects. Differentiating the two models was an interaction between the Time and Challenge variables (Time×Challenge) that was only present in the test model.

*Null model*: gene expression ~ Time + challenge + order + weights + individual + collection date + ε

*Test model*: gene expression ~ Time × challenge + order + weights + individual + collection date + ε

To identify genes with expression changes that occurred specifically during peanut but not placebo challenge, we used a likelihood ratio test (LRT) to test for a significant additional contribution of the interaction between time and peanut vs. placebo exposure on gene expression in the test model when compared to the null model. The LRT was implemented in R using the lmtest package (www.R-project.org).

A similar approach was applied to RNA-seq data generated from the replication cohort. However, because this cohort only included data from peanut challenges, the LRT was used to test for an effect of "Time".

**Leukocyte deconvolution analyses**. Raw RNA-seq read counts for genes used in lme model analysis ($n = 17,337$) from 107 samples were first normalized using DESeq2[70]. We used CIBERSORT[11] to estimate the proportions of leukocyte sub-populations in each RNA-seq sample from DESeq2-normalized gene counts, using the "L22" validated gene signature matrix;[11] estimates and associated P-values were based on 1000 permutations of the data, as implemented in the CIBERSORT pipeline. To be inclusive of all individual samples in our data set, no significance filter was applied to the estimated cell fractions. Following the same framework used for gene expression analysis, for each of the cell types estimated, an LRT was used to assess differences between null and test lme models, in which the test model again included an interaction effect.

*Null model*: cell fraction ~ Time + challenge + order + individual + collection date + ε

*Test model*: cell fraction ~ Time × challenge + order + individual + collection date + ε

P-values produced by the LRT were FDR corrected using the Benjamini-Hochberg method.

In the replication cohort, following the same methods for the inference of leukocyte fractions from RNA-seq data using CIBERSORT, the LRT was used to test for effects of "Time" on cell fraction, using equivalent models to those outlined above for the gene expression analysis.

**Coexpression network and gene ontology analyses**. WGCNA was conducted using the WGCNA and coexpp R packages (https://bitbucket.org/multiscale/coexpp). To construct a weighted gene coexpression network for food allergy, 139 RNA-seq samples from food allergic children were used, including the 107 samples from peanut allergic discovery cohort plus 32 samples from seven individuals with non-peanut food allergies who were recruited from the Mount Sinai Allergy/Immunology practice during the same study period and analogously sampled during oral food challenges to their suspected food allergen (Supplementary Table 5). The rationale for including samples from food allergic children more broadly and not just those from peanut allergic subjects was that our goal was to build a coexpression network for food allergy upon which we could then project our peanut genes. Prior to WGCNA, the same expression level thresholds used previously for the lme model analysis were applied to the data set, resulting in 17,328 genes. Gene counts were again normalized using edgeR/voom, and the effects of collection date were removed. The resulting residuals were used for weighted gene coexpression network construction. An adjacency matrix was generated by applying a power function (beta = 6; Supplementary Fig. 7) to a pair-wise gene correlation (Pearson's) matrix, initially estimated using per gene residual values. This was then transformed into a topological overlap matrix, from which module assignments were made using a dynamic tree cutting method.

We tested for enrichment of peanut genes in each module using the Fisher's exact test. Enrichments of coexpression modules for GO "Biological Process" terms were assessed using GOrilla;[16] the full set of genes input into WGCNA was used as background.

**Probabilistic causal gene network construction**. We constructed a directed probabilistic causal gene network by integrating peripheral blood gene expression data from 526 patients with IBD[20] and eQTL using RIMBAnet[19, 24, 25], a software package we developed for constructing integrative molecular Bayesian networks. As these networks do not scale linearly with increasing nodes, we took the top 25% of varying genes to build this network[19, 25]. Continuous data was used for calculating partial priors, which are then used as priors in the network construction. Additional priors included genes that are *cis* eQTLs. For the eQTL priors, if a gene also has a strong eQTL associated with it in *cis*, such a gene can be considered as a parent node, given the genotype cannot be the effect of a gene expression change. The data was discretized into three states for each gene: high expression levels, low expression levels, and unexpressed. This is done by first normalizing the values for

each gene to ensure a normal distribution. Then, k-means clustering ($k = 3$) is used with the option of dropping groups should there not be enough members to fill it to assign the values for each sample. In a case for which there are only two clusters they would be classified as high and low[19, 24, 25].

To support our use of the IBD network, a peanut allergy-specific replication causal network was constructed using data from the peanut allergic discovery and replication cohorts. Specifically, the same genes that were used as input for the IBD network that were also expressed in the peanut allergic cohorts ($n = 7055$) were used. The data were discretized in the same fashion as the IBD data with analogous use of RIMBAnet[19, 24, 25] to construct a probabilistic causal network.

**Key driver analysis**. Key driver analysis was conducted using the KDA package in R[17]. The package first defines a background subnetwork by looking for a neighborhood K-steps away from each node in the target gene list in the network. Then, stemming from each node in this subnetwork, it assesses the enrichment in its k-step (k varies from 1 to K) downstream neighborhood for the target gene list. In this analysis, we used $K = 7$. KDA was run on both the IBD network as well as the peanut allergy-specific causal network.

**Data availability**. Data for this study (doi:10.7303/syn10212437) is available via Synapse, a software platform for open, reproducible data-driven science. The direct link to the data is: https://www.synapse.org/#!Synapse:syn10212437/files/.

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

## Acknowledgements

We thank the participating subjects and their families who made this study possible. We thank Dr. Marshall Plaut, CoFAR scientific and medical officer and chief of the National Institute of Allergy and Infectious Diseases, Food Allergy, Atopic Dermatitis, and Allergic Mechanisms Section. We thank the staff of the clinical research units at each institution and the Statistical and Clinical Coordinating Center. We also thank the Mount Sinai Genomics Core and Scientific Computing at the Icahn School of Medicine at Mount Sinai. This study was supported by the National Institutes of Health (K08AI093538, R01AI118833, U19AI066738, and U01AI066560) and the Mindich Child Health and Development Institute at Mount Sinai. The project was also supported by grant nos. UL1 TR-000154 (National Jewish), UL1 TR-000067 (Mount Sinai), UL1 TR-000039 (Arkansas), UL1 TR-000083 (U North Carolina) and UL1 TR-000424 (Johns Hopkins) from the National Center for Research Resources (NCRR), a component of the National Institutes of Health.

## Author contributions

S.B. directed and designed the study with advisory input from E.E.S. S.B. coordinated the recruitment of subjects and sample collection with advisory input from H.A.S. H.A.S., S.S., R.W., A.W.B., S.M.J., and D.Y.M.L. enabled access to subjects and supervised the clinical research staff. A.G. and H.H. coordinated sample intake and processing. H.S. and C.T.W. performed RNA-seq preprocessing. P.D., A.H., and S.B. curated the clinical data. S.B., R.S.G., E.E.S., C.T.W., and A.T.C. designed the statistical and computational analyses. C.T.W., A.T.C., S.B., E.E.S., and Y.C. performed the statistical and computational analyses with advisory input from R.S.G., A.J.S., N.D.B., and G.H. C.T.W., A.T.C., and S.B. wrote the manuscript. All authors critically reviewed and edited the manuscript. All authors contributed significantly to the work presented in this paper.

## Additional information

**Competing interests:** The authors declare no competing financial interests.

