## [Peer Review File · Nature Communications]

Reviewer #1 (Remarks to the Author):

The authors have profiled the dynamic changes in whole blood transcriptomes during an induced acute allergic reaction to peanut in peanut allergic individuals in a controlled clinical setting. The study presents an excellent design combined with an in depth bioinformatics analysis of whole blood transcriptional changes, identifying key driver genes and key cell types that are altered during acute allergic reaction to peanut. The findings are novel and significant in the field of food allergy.

Major comments

The study is highly significant, yet the results are based on one cohort. The replication of the study with the same design in a second cohort or focusing specifically on the analysis of the altered cell subsets, e.g. neutrophils, in a second cohort would be highly desirable, however this is out of the scope of the current manuscript. Instead, the study could significantly benefit from replicating the key findings *ex vivo* or testing the transcriptional model with key driver genes and their downstream networks *in vitro*. The study is descriptive at current stage and some mechanistic *ex vivo* experiments on key cell types would greatly complement and functionalize the bioinformatics data and provide an insight into peanut-dependent transcriptional changes in key cell types (versus transcriptional changes reflecting the altered composition of leukocytes in whole blood).

For example, the whole blood from allergic individuals could be exposed to peanut extract *ex vivo* (the design mimicking the clinical study) and the key cell types could be subsequently analysed, using e.g. the transcriptomic and protein analyses of key driver genes and their gene networks or signalling pathways (NF- κ B), mass cytometry profiling of cellular subsets, functional experiments in key cell types, e.g. neutrophil chemotaxis on small volumes of whole blood using microfluidic platforms (see C.N. Jones et al. *J Vis Exp.* 2014; Hamza B. & Irimia D. *Lab Chip.* 2015). The possible confounding effects of anti-histaminic drugs and epinephrine that cannot be avoided in the clinical study could be also explored.

The isolation of intact neutrophils from peripheral blood and *in vitro* cell culturing for e.g. loss of function studies of key driver genes and downstream networks could be an alternative option (e.g. see Zimmermann M et al. *Nat Commun.* 2015;6:6061), however it might prove technically challenging.

Minor comments

- The authors should explain the rationale (clinical, organizational etc.) for selecting the 2hr and 4hr time points as a readout points for transcriptome analyses during the acute peanut allergic reaction. The development of the allergic reaction development in 19 subjects should be described over the analysed time points (0hr, 2hr and 4hr during the peanut challenge), so that the reader can align the clinical course with dynamics of transcriptome changes. Some brief background on the development of allergic reaction – cell types involved and basic molecular mechanisms, should be provided in the introduction for a broader readership.
- Why the activated mast cells and basophils were not included in the leukocyte deconvolution analysis considering their key role in acute peanut allergic reactions?
- Leukocyte deconvolution analysis on whole blood transcriptomes identified macrophages M0 as a cell type with significant changes during the acute peanut allergic reaction. Could the authors discuss these result considering that macrophages are primarily tissue resident cells?
- The antihistaminic drugs are a cornerstone in treatment acute allergic reactions and not using them would be highly unethical. All 19 studied subjects received antihistaminic drugs during the peanut challenge (but probably not the placebo challenge) suggesting that antihistaminics might

induce or mask some of the transcriptome changes in the whole blood. Could the authors comment on this point?

- Based on the study design, 114 samples should be analysed, but the authors report that 107 sample were analysed. Why were the remaining 7 samples excluded?
- Did any subject report any adverse reactions during the placebo challenge? The study subjects were blinded and randomised and the authors took into account the order of peanut-placebo challenge when identifying the peanut genes. Still, could any potential bias occur during the placebo challenge (e.g. distress) in subjects that received peanut challenge first? Can the authors comment on this.
- How was the randomisation of patients with regard to the order of-peanut challenge done? The interventions should be described in sufficient details to allow the repetition of a study by other researchers.

Reviewer #2 (Remarks to the Author):

This is an interesting paper from an expert group, using innovative analyses of blood transcriptomics from a carefully conducted clinical challenge to investigate oral peanut allergy. There is very little previous transcriptomic data on oral allergy in the published literature.

Methodologically, the clinical study appears well thought out and well controlled. It is safe to assume the RNA sequencing and initial downstream analyses are state of the art. The identification of cell subsets using CIBERSORT is convincing. The Key Driver Analysis is harder to assess without some indication of the robustness of the results.

A major problem is that the list of genes and networks that arise from the analysis is difficult to interpret biologically.

Interpretation would be helped by an additional column in table 2 to summarise the known function or origin of each gene. PLP2, for example may have a role in colonic epithelium, CD22 is a B-cell marker, DGKG is found in macrophages, BST1 in pre B-cells and so on.

In any case the list of genes is a surprise, and does not contain any of the usual suspects for cellular responses to allergen challenge. This may well be because non-one has looked in any detail at oral allergen challenge previously, but given the complexity of the analyses it feels as if the paper really needs some form of direct biological replication of the results. I understand that these data are not available in the published literature, and the arguments for using data from IBD are not convincing. In the same vein, actual cell counts from a Coulter counter would add weight and reassurance to the bioinformatic results.

Line 133: Given the number of genes under consideration, and the relatively small sample size, the use of a statistical stringency less than the Bonferroni threshold may cause problems. What happens if the stringent threshold is applied?

The WGCNA analysis is interesting, and appears very robust. The blue peanut module seems likely to contain the most important information from the study. Unfortunately I could not find Supplementary Table 3 in the submitted documents. A chord diagram might help in showing the intersection between the module(s) and the peanut genes. Schadt and others have shown consistently that WGCNA modules correspond to particular cell types, and some form of attribution of the peanut module to the (inferred) cell types would be particularly helpful.

The KDA analysis has not been used widely in the literature, and it is very difficult to interpret the validity of the findings in this paper. Here again some form of validation would appear necessary to accept these results.

Reviewer #3 (Remarks to the Author):

Integrative transcriptomic analysis of serial peripheral blood samples of children allergic to peanuts was used to define causal genes and pathways. Cohort of 19 children undergoing double-blind, placebo-controlled oral food challenges was used.

Overall, authors should remove statements about causal genes affecting food allergies – there are different food allergies and the current paper only studies peanut allergies – so these genes cannot (yet) be generalized to all food allergies.

Considering a huge range of reaction to peanuts of individual subjects – one has to wonder how the outliers on both ends affect the results. It is unclear why only 6 samples per time point/group – there should be 9 and 10.

Top 30 identified genes – was the 31st gene with $pvalue > 0.01$?

“the gene expression levels for six selected peanut genes are plotted” – how selected and why specifically 6?

“Employing a leukocyte cell-type deconvolution algorithm,¹⁰ we inferred the proportions of 19 leukocyte populations” – could work well under specific conditions – but varying conditions (in this case peanut allergy response) could make the algorithm inaccurate. Was it ever validated on the same data/analysis?

“we next performed gene ontology (GO) enrichment analysis. This revealed significant 195 enrichments of the peanut response module for inflammatory pathways” - since GO enrichment analysis was done – authors identified biological processes, molecular function (possibly localization) – not pathways. Pathway enrichment analysis would need to be done to determine enriched pathways.

It is not justified why the data with 526 IBD subjects is the best for constructing the Bayes net for this study. It is also not justified why an arbitrary top 7200 genes were selected. Statistics should be used to guide the cut offs. “We reasoned that there is some overlap 234 between the two conditions with regard to chronic immune dysregulation and pro-inflammatory 235 processes such that major edges of the underlying disease network architecture in IBD could 236 inform gene coregulation relevant to inflammatory response to peanut” – it would be useful to provide some data and analysis to substantiate “the reason” – there are many conditions that relate to immune system – yet each will overlap with peanut allergy response differently. Using IBD as an underlying model provides bias that is not characterized, and thus results from the analysis cannot be properly interpreted. The statistics provided as a support for the argument would need more description about what was used as a background in the comparisons. Node overlap is one side of the comparison – considering both IBD and peanut allergy relate to immune system – it is not surprising to see the high overlap. But there is no evidence that the edges would be conserved.

“The data that support the findings of this study will be made publicly available upon completion 546 of the CoFAR6 clinical trial of epicutaneous immunotherapy for peanut allergy.” – the data would need to be made publicly available with the publication – to support open science.

Making a statement that sometime in the future data will be made available is inappropriate. Especially, authors highlight the lack of omics data sets on allergic response – the paper should

make the data available.

Figure 4 – using grey font on white background reduces contrast. Panel B – font quite small on the x axis. Panels C/D – forced image view – more information would be conveyed if a table is presented. Provides no scale, font varied too much.

Figure 5 provides an interesting view – but considering the overlap of nodes and edges – does not convey much information.

all results should be corrected for age, allergy severity and sex.

Reviewers' comments:

Reviewer #1 (Remarks to the Author):

Comment 1: “The authors have profiled the dynamic changes in whole blood transcriptomes during an induced acute allergic reaction to peanut in peanut allergic individuals in a controlled clinical setting. The study presents an excellent design combined with an in-depth bioinformatics analysis of whole blood transcriptional changes, identifying key driver genes and key cell types that are altered during acute allergic reaction to peanut. The findings are novel and significant in the field of food allergy.”

Major comments

“The study is highly significant, yet the results are based on one cohort. **The replication of the study with the same design in a second cohort or focusing specifically on the analysis of the altered cell subsets, e.g. neutrophils, in a second cohort would be highly desirable, however this is out of the scope of the current manuscript.** Instead, the study could significantly benefit from replicating the key findings *ex vivo* or testing the transcriptional model with key driver genes and their downstream networks *in vitro*. The study is descriptive at current stage and some mechanistic *ex vivo* experiments on key cell types would greatly complement and functionalize the bioinformatics data and provide an insight into peanut-dependent transcriptional changes in key cell types (versus transcriptional changes reflecting the altered composition of leukocytes in whole blood).”

“For example, the whole blood from allergic individuals could be exposed to peanut extract *ex vivo* (the design mimicking the clinical study) and the key cell types could be subsequently analyzed, using e.g. the transcriptomic and protein analyses of key driver genes and their gene networks or signaling pathways (NF- κ B), mass cytometry profiling of cellular subsets, functional experiments in key cell types, e.g. neutrophil chemotaxis on small volumes of whole blood using microfluidic platforms (see C.N. Jones et al. *J Vis Exp.* 2014; Hamza B. & Irimia D. *Lab Chip.* 2015). The possible confounding effects of anti-histaminic drugs and epinephrine that cannot be avoided in the clinical study could be also explored.”

“The isolation of intact neutrophils from peripheral blood and *in vitro* cell culturing for e.g. loss of function studies of key driver genes and downstream networks could be an alternative option (e.g. see Zimmermann M et al. *Nat Commun.* 2015;6:6061), however it might prove technically challenging.”

>>> We thank Reviewer 1 for their positive comments and suggestions regarding the work reported in our initial submission. We feel that several of the suggested *ex vivo* experiments are outside the scope of this particular project and more importantly, may not accurately model the novel *in vivo* findings we report. However, regarding the reviewer’s first suggestion of “replication of the study with the same design in a second cohort” as “highly desirable,” we are pleased to report that we have done exactly that for this resubmission in direct response to this comment. Specifically, we have replicated our main findings in a second cohort of 21 independent peanut allergic children undergoing peanut oral food challenges with serial profiling and analysis of their whole blood transcriptomes. The results of these replication analyses are provided in a **new section of Results: “Replication of gene expression and leukocyte subset signatures in an independent cohort of 21 peanut allergic children undergoing peanut challenge,”** updated **Tables 1 and 2**, and **new Supplementary Figures 4 and 5** of the

edited manuscript. We found very strong concordance between the discovery and replication cohort results for the gene expression and leukocyte deconvolution analyses, as well as support for our causal network and key driver results with this second cohort (**Results lines 216-236, and lines 341-353** of the marked version). For ease, steps for which replication analyses were conducted are indicated by maroon arrows in updated **Figure 1**.

Minor comments

Comment 2: “The authors should explain the rationale (clinical, organizational etc.) for selecting the 2hr and 4hr time points as a readout points for transcriptome analyses during the acute peanut allergic reaction. The development of the allergic reaction development in 19 subjects should be described over the analysed time points (0hr, 2hr and 4hr during the peanut challenge), so that the reader can align the clinical course with dynamics of transcriptome changes. Some brief background on the development of allergic reaction - cell types involved and basic molecular mechanisms, should be provided in the introduction for a broader readership.”

>>>We thank Reviewer 1 for this helpful comment and have added information to the manuscript on the suggested points as follows:

In Methods, lines 587-593, we now state:

“Fixed collection time points were chosen to ensure uniformity in collection strategy across peanut and placebo challenges performed by varying blinded study personnel at different sites. The rationale for sampling at 2 hours is that experience at our study sites supports that most children who react during a peanut challenge exhibit symptoms within the first 2 hours. We additionally sampled at 4 hours because we hypothesized that adaptive immune responses could take longer to manifest as gene expression changes, and 4 hours is a typical length of time for a food challenge visit to conclude.”

Additionally, in Results line 123 we now state:

“All children reacted within the first 2 hours of the peanut challenge.”

In response to Reviewer 1’s suggestion to include brief background on the development of allergic reactions for the broader readership, we now state in Introduction, lines 59-64:

“The development of peanut allergy involves deviation from mucosal and cutaneous immune tolerance, such that dietary antigens presented by antigen presenting cells lead to an adverse Th2-cell skewed response, priming of innate immune effector cells, and alteration of cytokine milieu such that subsequent food allergen-specific antigen exposure leads to IgE-mediated acute reactions.³⁴”

Comment 3: “Why the activated mast cells and basophils were not included in the leukocyte deconvolution analysis considering their key role in acute peanut allergic reactions?”

>>> The leukocyte deconvolution analysis is constrained by cell subsets included in the CIBERSORT algorithm. Basophils are among the least abundant cells in peripheral blood, and the fraction of basophils is not estimated by the current CIBERSORT algorithm. In contrast, activated mast cells are included in the algorithm; however, these were estimated to represent 0% of the cells in the samples. Resting mast cells, were observed at relatively low frequencies in our samples (mean, ~1%), and no significant changes were observed during the peanut vs.

placebo challenge (please see **Fig 3**). This apparent lack of association could result from their low estimated fractions in peripheral blood.

Comment 4: “Leukocyte deconvolution analysis on whole blood transcriptomes identified macrophages M0 as a cell type with significant changes during the acute peanut allergic reaction. Could the authors discuss these result considering that macrophages are primarily tissue resident cells?”

>>> We agree that macrophages are typically considered tissue resident cells, and this does appear to be reflected in the CIBERSORT estimates, which suggest that the average macrophage cell fractions are quite low. For M0 cells, specifically, the mean is ~2.5%, whereas committed M1 and M2 lineages are estimated at mean fractions of ~.02% and 0%, respectively. Monocytes, in contrast, have a mean fraction of ~22%. Perhaps the elevation of M0 fractions during acute peanut reactions reflects increases in macrophage migration to sites of inflammation. For example, increases in blood monocytes and macrophages has been noted during inflammation associated with Kawasaki disease (PMID:16045726, PMID:11350607).

In the replication cohort now included in the revised version of this manuscript, we found that the exact same 3 cell types were significantly associated with time during the peanut allergic reasons and in the same direction as in the discovery cohort (see new **Results** section: “**Replication of gene expression and leukocyte subset signatures in an independent cohort of 21 peanut allergic children undergoing peanut challenge**” and new **Supplementary Figure 5**). The fact that we replicated this association in a second cohort gives us confidence that we have identified a genuine signature.

Additional studies to further validate our findings and better understand the potential functions of this particular cell type are certainly warranted as we state in the **Discussion, lines 484-486**: “Our study provides data on 19 inferred leukocyte cell fractions (Fig. 3), highlighting three cell subsets that could be directly profiled in future studies.”

Comment 5: “The antihistaminic drugs are a cornerstone in treatment acute allergic reactions and not using them would be highly unethical. All 19 studied subjects received antihistaminic drugs during the peanut challenge (but probably not the placebo challenge) suggesting that antihistaminics might induce or mask some of the transcriptome changes in the whole blood. Could the authors comment on this point?”

>>> We recognize this issue, as we did for the potential impacts of epinephrine on gene expression, which is why we performed a secondary analysis stratified by epinephrine use (**Supplementary Figure 3**), given that a subset of participants did not receive epinephrine. We agree with Reviewer 1 that antihistamine receipt by all participants during peanut challenge presents a challenge, not just for our study, but for all studies seeking to characterize *in vivo* allergic reactions in human subjects. Given the ethical constraints that preclude withholding treatment, teasing potential drug effects is an extremely difficult to impossible issue to resolve in human studies. That said, we note that many of the broader patterns of gene expression observed suggest the initiation of inflammation. If anything, antihistamine effects might lead to false negatives rather than false positives. Although we cannot directly delineate antihistamine effects from this study, we would not expect these to lead to the broader inflammatory signatures observed.

Additionally, we checked the Connectivity MAP (CMAP) (PMID: 17008526, 17186018) a Broad-Institute-based collection of genome-wide transcriptional expression data from cultured human cells treated with 1039 drugs and bioactive molecules. We pulled gene expression data from experiments involving the antihistamines diphenhydramine and cetirizine. CMAP releases the ranking of genes in each experiment based on the ratio of cell line with drug to cell line without drug. Using these experimental data, we defined 45 antihistamine signature gene sets based on varying thresholds (i.e. the top 100, 500, 1000, 2000 and 5000 genes differentially expressed in each of 9 cell lines treated vs. not treated with antihistamine). We tested for enrichment of each of these antihistamine signature sets with the peanut gene set using a permutation testing approach with 10,000 permutations. In 43 of the 45 tests run, the overlap between antihistamine signature and our peanut gene set was not significant. This supports that antihistamine effects on gene expression did not underlie the peanut genes identified.

Comment 6: “Based on the study design, 114 samples should be analysed, but the authors report that 107 sample were analysed. Why were the remaining 7 samples excluded?”

>>> We have clarified this point in the **Results section (lines 142-144)**. The design of the study was to collect 6 samples per subject (samples at baseline, 2 hours and 4 hours each from the peanut and placebo challenges). While this was fully executed in 17 of the 19 subjects, only peanut challenge-related samples were collected by the study staff in 2 of the subjects. Among the 108 samples collected and then sequenced, a single outlier was identified and removed during our quality control checks of the RNA-seq data (see **Methods line 627-628** and **Supplementary Figure 1**), resulting in a total of 107 samples for analysis. The statistical models used in this study-- linear mixed effects models -- handle missing data robustly with maximum likelihood estimation.

In Results lines 142-144, we now state;

“Six samples were collected from each of the 19 subjects, except for 2 participants, for whom samples were obtained during peanut challenge only, resulting in a total of 108 samples collected.”

Regarding the removal of the outlier, in lines 627-628 we state:

“Multidimensional scaling analysis was used to identify outlier RNA-seq samples, resulting in removal of 1 sample (Supplementary Figure 1) from the discovery cohort.”

Comment 7: “Did any subject report any adverse reactions during the placebo challenge?”

>>> No, none of the subjects reported adverse reactions during the placebo challenge.

In Results, lines 118-119, we now state:

“None of these subjects reported symptoms during placebo challenge.”

Comment 8: “The study subjects were blinded and randomised and the authors took into account the order of peanut-placebo challenge when identifying the peanut genes. Still, could any potential bias occur during the placebo challenge (e.g. distress) in subjects that received peanut challenge first? Can the authors comment on this?”

>>> It was our intention to mitigate any bias by conducting this trial in a double blinded and randomized fashion. The order of the peanut and placebo challenges was randomized, and both

participants and study staff were blinded to the challenge food. Ten of the 19 subjects (52%) underwent the peanut challenge first, thus neither substance was over-selected as the first challenge. To be certain, we additionally corrected for challenge order in the statistical model to account for any potential residual bias (see **Methods, lines 636-648**). Further, none of the subjects reported distress or any other symptom during the placebo challenge. Each of the symptoms listed in **Table 1** was explicitly evaluated for during the placebo challenges.

Comment 9: “How was the randomisation of patients with regard to the order of-peanut challenge done? The interventions should be described in sufficient details to allow the repetition of a study by other researchers.”

>>> Order of peanut vs. placebo challenge was determined by coin flip by blinded staff, the most common method of simple randomization (PMID 21772732). This resulted in balanced randomization, with 10 of the 19 subjects (52%) undergoing peanut challenge first.

To add this detail on randomization method, in Methods, line 578 we now state;
“*Following randomization by coin flip and under close medical supervision with blinding of subjects and staff, each subject ingested incremental amounts of peanut at 20 minute intervals until allergic reaction or until final cumulative dose of 1.044 grams protein ingested on one day, and ingestion of equivalent incremental doses of placebo oat powder in similar fashion on another day shortly before or after.*”

Reviewer #2 (Remarks to the Author):

Comment 1: “This is an interesting paper from an expert group, using innovative analyses of blood transcriptomics from a carefully conducted clinical challenge to investigate oral peanut allergy. There is very little previous transcriptomic data on oral allergy in the published literature.”

“Methodologically, the clinical study appears well thought out and well controlled. It is safe to assume the RNA sequencing and initial downstream analyses are state of the art. The identification of cell subsets using CIBERSORT is convincing. The Key Driver Analysis is harder to assess without some indication of the robustness of the results.”

“A major problem is that the list of genes and networks that arise from the analysis is difficult to interpret biologically. Interpretation would be helped by an additional column in table 2 to summarise the known function or origin of each gene. PLP2, for example may have a role in colonic epithelium, CD22 is a B-cell marker, DGKG is found in macrophages, BST1 in pre B-cells and so on.”

>>> We thank Reviewer 2 for their thoughtful review of our manuscript. We hope that our revisions capture the concerns and suggestions outlined in your comments.

Specifically, with respect to Reviewer 2’s comment on a lack of gene-level information for the list of genes in Table 2, we have included basic functional annotations for each of these genes in a new table now included as **Supplementary Table 1**. The information is based on literature surveys for each gene, and supporting references are provided in the supplementary material. We fully address Reviewer 2’s comment regarding Key Driver Analysis in the next response.

Comment 2: “In any case the list of genes is a surprise, and does not contain any of the usual suspects for cellular responses to allergen challenge. This may well be because non-one has looked in any detail at oral allergen challenge previously, but given the complexity of the analyses it feels as if the paper really needs some form of **direct biological replication of the results.** I understand that these data are not available in the published literature, and the arguments for using data from IBD are not convincing. In the same vein, actual cell counts from a Coulter counter would add weight and reassurance to the bioinformatic results.”

>>> First, with regard to biological replication, we now report replication of our findings in a second cohort of 21 peanut allergic subjects undergoing peanut challenges with serial whole blood transcriptome profiling. We are pleased to report that the results regarding gene expression changes and cell subset changes are highly concordant between the discovery and replication cohorts (please see our **response to Reviewer 1, Comment 1** as well as new **Results** section: “***Replication of gene expression and leukocyte subset signatures in an independent cohort of 21 peanut allergic children undergoing peanut challenge,***” Tables 1 and 2, **Supplementary Figures 4 and 5**, and **Results lines 216-236, and lines 341-353**

Second, in response to concern about our use of data from an IBD cohort, for this revision **we constructed a probabilistic causal gene network using combined data from the discovery and new replication cohort of peanut allergic children (see Methods, lines 728-733), finding strong support for the IBD network.** As background on our overall approach, we were motivated to use the IBD cohort for construction of the probabilistic causal network because this cohort contains many samples (N=526) as well as genetic prior information (expression quantitative trait loci (eQTL)), thus enabling construction of a robust and informative causal network (PMID: 15237224 18552845 22509135 17432931) that simply was not possible with the peanut allergy discovery cohort. Constructing a probabilistic causal network is an NP-hard problem. As such, it requires many heuristic approaches to search and find networks that represent the data. It is possible to identify networks that appear to fit the data optimally, but which are not the best fitting network (in technical terms, the optimization problem run to find the best network gets trapped in a local maximum). With greater power (i.e. larger sample size) and prior information, such as eQTLs, we reduce the risk of getting trapped in a local maximum, thus ensuring that we reach the best fitting network. A network constructed with the IBD cohort was therefore the best option available.

However, in response to this comment and to validate the IBD network’s relevance to peanut allergy, we leveraged the new replication cohort by combining these data with the discovery cohort to build a peanut allergy-specific probabilistic causal network (see **Methods, line 728-733**). We note that the peanut causal network is not as robust as the IBD network given the still limited sample size for its construction and lack of eQTLs available as prior data for input. eQTL data serve to constrain the size of the search space by enhancing the ability to directly identify causal relationships among genes (PMID 15965475, 22509135). This peanut casual network was therefore constructed as a validation of the IBD network, which is indeed what we observed from it. When we ran the same subnetwork analysis but in the peanut network, the overlap between the IBD and peanut subnetworks was 3-fold enriched over what would be expected by chance (Fisher’s exact test $P= 3.3 \times 10^{-224}$). Further, key driver analysis of the peanut-allergy causal network revealed a highly significant overlap with key drivers learned from the IBD network (OR: 16.8, Fisher’s Exact $P= 2.4 \times 10^{-7}$), thus providing support for our findings from the original network.

In Results, lines 341-353, we now state:

“To ensure that the IBD network was capturing biology relevant to peanut allergy, we constructed a validation causal gene network using the peanut allergic discovery and validation cohorts (**Table 1**). As context, we used the IBD network as our discovery causal network because it could be built on a much larger dataset (n=526) with genetic prior information (expression quantitative trait loci (eQTL)), thus enabling construction of a robust and informative causal network.^{21, 28-30} The peanut causal network we built is not as robust as the IBD network given the limited sample size for its construction and lack of eQTLs available as input, and was therefore used to validate findings from the IBD network. Indeed, when we ran the same subnetwork analysis as above but in the peanut network, the overlap between the IBD and peanut subnetworks was 3-fold enriched over what would be expected by chance (Fisher’s exact test $P= 3.3 \times 10^{-224}$). Further, KDA of the peanut allergy-specific causal network identified key drivers also significantly overlapping with those from the IBD network (OR 16.8, Fisher’s exact test $P= 2.4 \times 10^{-7}$), thus lending confidence to our causal network findings.”

Finally, with respect to the suggestion to obtain direct cell counts in our study cohort, we specifically took the CIBERSORT deconvolution approach because we were limited by the small volumes of blood that could be collected serially from pediatric subjects with IRB approval. Considering our study design required 6 collections per child during the peanut and placebo challenges, we collected 2.5mL per time point, which was sufficient for RNA isolation and RNA-sequencing, but not additionally sufficient for flow cytometry-based assays for multiple cell populations and/or Coulter counter. We recognize the value of direct cell counts however, and state in **Discussion lines 468-486**: “Our leukocyte deconvolution approach was unbiased so as to inclusively capture cellular expression signatures across peripheral blood. Due to limitations on the volume of blood that can be collected from children, especially in the context of serial sampling, flow cytometry-based assays to target multiple cell populations at each time point was not feasible. Our study provides data on 19 inferred leukocyte cell fractions (**Fig. 3**), highlighting three cell subsets that could be directly profiled in future studies.”

Comment 3: “Line 133: Given the number of genes under consideration, and the relatively small sample size, the use of a statistical stringency less than the Bonferroni threshold may cause problems. What happens if the stringent threshold is applied?”

>>> We described our results in two ways. First, we applied a Bonferroni correction to our gene lists to identify genes exhibiting the most significant changes in expression during the peanut challenge relative to placebo challenge. Indeed, these 30 genes were highly replicated in our newly reported replication cohort (**Table 2**). However, the number of genes in this set (n=30) is too small for downstream analyses that are based on testing gene set overlaps and enrichments. Therefore, we used a less stringent statistical threshold of $P < 0.005$ to define peanut genes for downstream analyses (**Supplementary Table 2**). That said, $P < 0.005$ corresponds to an FDR of 0.0398, supporting that the false discovery rate from this relatively less stringent cutoff is still minimal.

Comment 4: “The WGCNA analysis is interesting, and appears very robust. The blue peanut module seems likely to contain the most important information from the study. **Unfortunately I could not find Supplementary Table 3** in the submitted documents. A chord diagram might help in showing the intersection between the module(s) and the peanut genes. Schadt and others have shown consistently that WGCNA modules correspond to particular cell types, and some form of attribution of the peanut module to the (inferred) cell types would be particularly helpful.”

>>> We are sorry to learn that Reviewer 2 was not able to access **Supplementary Table 3** (now **Supplementary Table 4**). From our records of the submitted documents, it should have been available. We will ensure that it is not missing in our resubmission. Relevant to this comment, **Supplementary Table 4** does show the number of member genes in each module, as well as the number of peanut genes in each model. Regarding the intersection between modules, each gene is exclusively assigned to a single module, and thus there are no gene overlaps between them.

With respect to Reviewer 2's second point, we agree that it would be interesting to be able to directly overlap coexpression module members to cell-type specific gene sets. However, at present, it is not apparent to us the methodology for such an analysis based on the data we have. The cell subset inference done by CIBERSORT is based on collective expression signatures from a limited subset of genes. Each gene is not necessarily assigned to or representative of any one given cell type, and thus we are unable to directly test for an enrichment of modules in "cell-specific" gene subsets.

Comment 5: "The KDA analysis has not been used widely in the literature, and it is very difficult to interpret the validity of the findings in this paper. Here again some form of validation would appear necessary to accept these results."

>>> As described in our response to **Reviewer 2, Comment 2** and directly responsive to this comment, we used the discovery cohort and new replication cohort to construct a peanut allergy-specific probabilistic causal gene network, which had not been possible before. We performed key driver analysis on this new network and found that the key drivers identified from the peanut-specific network significantly overlapped with those identified in our key driver analysis of the network from our initial submission (OR 16.8, Fisher exact test P-value = 2.4×10^{-07}).

Additionally, while KDA is a relatively newer form of network analysis, its use has been in place for a decade. KDA-like analyses have been developed and applied in dozens of papers from leaders in the network biology field, including the work of Andrea Califano at Columbia University, Aviv Regev at the Broad Institute, Trey Ideker at UCSD, Richard Bonneau at NYU, and others. In addition there are DREAM competitions that have been specifically designed to calibrate and validate the many methods developed for KDA. Finally, with the KDA method we have applied in our manuscript, a number of papers have been published that support the approach as robust and leading to novel biological discovery (PMIDs: 22806142, 23622250, 27896968, 19741703, 18358334, 22509135). **We have added a selection of these references to Results, lines 326-328 for the readership, where we now state:**

"Key drivers have been found to be reproducible and successfully validated in different contexts as regulating genes of interest.²⁵⁻²⁷"

Reviewer #3 (Remarks to the Author):

Comment 1: "Integrative transcriptomic analysis of serial peripheral blood samples of children allergic to peanuts was used to define causal genes and pathways. Cohort of 19 children undergoing double-blind, placebo-controlled oral food challenges was used."

“Overall, authors should remove statements about causal genes affecting food allergies - there are different food allergies and the current paper only studies peanut allergies - so these genes cannot (yet) be generalized to all food allergies.”

>>> First, we would like to thank Reviewer 3 for their thoughtful review our manuscript. We hope our responses below effectively address their questions/concerns.

We apologize if the wording in our initial submission suggested that our findings were generalizable to all food allergies and have carefully reviewed where we discuss peanut vs. food allergy more generally. We recognize that our study is focused on samples obtained from subjects with peanut allergy, and thus may not necessarily be relevant to other food allergies. Indeed, we completely agree that to understand what aspects of the signatures we have observed here are generalizable, or are specific to peanut, will require further study. We apologize if we implied otherwise.

For example, we explicitly attempted to address this point in our Discussion section, where we state in **lines 413-414**: “Investigation in cohorts with other food allergies will be required to determine the extent to which expression of peanut genes are specific to the allergic response to peanut.”

If Reviewer 3 is concerned about any statement in particular, we would be happy to edit it as needed.

Comment 2: “Considering a huge range of reaction to peanuts of individual subjects – one has to wonder how the outliers on both ends affect the results. It is unclear why only 6 samples per time point/group - there should be 9 and 10.”

>>> We are sorry if our study design was not clear. As we tried to illustrate in **Figure 1** and describe in Methods, the study design was that each participant would undergo double-blind, placebo-controlled peanut challenges with samples collected at baseline, 2 hours, and 4 hours each during the peanut as well as placebo challenges. Thus, this would result in 6 samples per subject. With 19 participants, one would then expect 19 samples per time point or 57 samples per group (e.g. peanut challenge samples vs. placebo challenge samples). We are not clear on why one would expect 9 or 10 subjects per time point or group as commented.

To additionally clarify the study design, we have added to the legend of Figure 1:

“For each subject, whole blood samples at baseline, 2 hours, and 4 hours were each collected during the peanut and placebo challenges.”

Regarding outliers, we show in **Supplementary Figure 1** that we did detect 1 outlier in our QC of the RNAseq data. This sample was removed from further analysis as described in **Methods lines 627-628**. We note that this study was not designed to address the potential impacts of response severity, although we have plans for future work that will investigate this.

Comment 3: “Top 30 identified genes - was the 31st gene with pvalue>0.01?”

>>> We showed the 30 genes with most significant P-value in **Table 2**, and these genes all had Bonferroni-corrected $P < 0.01$. We included a more extensive list of results in **Supplementary Table 2**. As this supplementary table shows, Reviewer 3 is correct that the 31st gene had

Bonferroni-corrected $P > 0.01$ ($P = 5.95 \times 10^{-07}$ and Bonferroni-corrected $P = 1.03 \times 10^{-02}$ to be exact).

Comment 4: "the gene expression levels for six selected peanut genes are plotted" - how selected and why specifically 6?"

>>> We have revised the text to explain the presentation of these specific 6 genes:

In Results (lines 164-167) we now state:

"The gene expression levels for six selected peanut genes are plotted in **Fig. 2** as examples of the expression pattern observed for genes identified by lme modeling; these six genes are the six key drivers identified by the downstream causal network analyses that follow. For context, plots for the top 30 peanut genes are provided in **Supplementary Fig. 2.**"

As noted, plots for the top 30 peanut genes are provided in **Supplementary Figure 2** for interested readers.

Comment 5: "Employing a leukocyte cell-type deconvolution algorithm,10 we inferred the proportions of 19 leukocyte populations" - could work well under specific conditions - but varying conditions (in this case peanut allergy response) could make the algorithm inaccurate. Was it ever validated on the same data/analysis?"

>>> CIBERSORT has been applied in other contexts, e.g., cancer (see PMC4852857). We would not expect "varying conditions" to confound our results, as leukocyte deconvolution was performed on each RNA-seq sample individually, before using the linear mixed effects model, so variation that occurs within individuals across time points or challenges would not impact the accuracy of each independent deconvolution. In addition, (please also see our responses to Reviewer 1, Comment 4 and Reviewer 2, Comment 2) we replicated the findings from our CIBERSORT analysis in a second independent cohort, which is now reported on in our revised manuscript (see **Results** section: "**Replication of gene expression and leukocyte subset signatures in an independent pediatric cohort of allergic individuals following peanut challenge**" and new **Supplementary Figure 5**).

Comment 6: "we next performed gene ontology (GO) enrichment analysis. This revealed significant 195 enrichments of the peanut response module for inflammatory pathways" - since GO enrichment analysis was done - authors identified biological processes, molecular function (possibly localization) – not pathways. Pathway enrichment analysis would need to be done to determine enriched pathways.

>>> We acknowledge that GO terms can have many different classifications, or rather, represent many different types of classes/concepts, and that those grouped under the broader classification of "biological process" may not explicitly represent pathways.

In response to this comment, we have modified the sentence to now read (lines 254-255):

"This revealed significant enrichments of peanut response module genes for many GO terms associated with inflammatory processes,..."

Additionally, we have carefully revised language throughout the paper to use “processes” where this would be more accurate.

Comment 7: “It is not justified why the data with 526 IBD subjects is the best for constructing the Bayes net for this study. It is also not justified why an arbitrary top 7200 genes were selected. Statistics should be used to guide the cut offs. “We reasoned that there is some overlap 234 between the two conditions with regard to chronic immune dysregulation and pro-inflammatory 235 processes such that major edges of the underlying disease network architecture in IBD could 236 inform gene coregulation relevant to inflammatory response to peanut” - it would be useful to provide some data and analysis to substantiate “the reason” - there are many conditions that relate to immune system - yet each will overlap with peanut allergy response differently. Using IBD as an underlying model provides bias that is not characterized, and thus results from the analysis cannot be properly interpreted. The statistics provided as a support for the argument would need more description about what was used as a background in the comparisons. Node overlap is one side of the comparison - considering both IBD and peanut allergy relate to immune system - it is not surprising to see the high overlap. But there is no evidence that the edges would be conserved.”

In response to this comment, we constructed a peanut allergy-specific Bayesian network that provided strong evidence in support of our results from the IBD network. **Please see our response to Reviewer 2, Comment 2 for details.**

Additionally, with regards to genes selected for the causal network construction, this was not done arbitrarily. The genes used to construct the network were chosen based on variance, taking the top 25% of all genes expressed. We used this same criterion when constructing the new peanut allergy-specific network now discussed in the revised manuscript. We note that this criterion has been used in other work as well, which we now cite as references.

In Methods, lines 711-713, we now state:

“As these networks do not scale linearly with increasing nodes, we took the top 25% of varying genes to build this network.^{21, 30”}

Regarding Reviewer 3’s comment about edge conservation, we specifically inferred topological information (i.e. key drivers) from the networks, which are more conserved and biologically meaningful than conservation of specific edges (PMID 22806142, 23622250, 27896968, 19741703, 18358334, 22509135). We found that key drivers of the IBD network highly overlapped with key drivers of the peanut allergy-specific network that we constructed to validate the IBD network (OR: 16.8, Fisher’s Exact P= 2.4×10^{-7}) (please see **Results lines 351-353**). This provides evidence that independent of edge conservation, the topologies of the IBD and peanut-allergy specific network are highly similar. We have additionally addressed conserved topology in **Results line 305**.

Comment 8: “The data that support the findings of this study will be made publicly available upon completion 546 of the CoFAR6 clinical trial of epicutaneous immunotherapy for peanut allergy.” - the data would need to be made publicly available with the publication - to support open science. Making a statement that sometime in the future data will be made available is inappropriate. Especially, authors highlight the lack of omics data sets on allergic response - the paper should make the data available.”

>>> We appreciate the reviewer's concern on this matter. We have discussed this issue with the Consortium for Food Allergy Research (CoFAR), with whom we collaborated on this study. We have now received permission to make these data publically available upon publication of our article. Upon publication, these data will be made publically available via Synapse under the study ID, doi:10.7303/syn10212437. The direct link is: <https://www.synapse.org/#!/Synapse:syn10212437/files/>. We had not been able to do this with the initial submission given the consortium's concern that sharing of data might lead to unintentional unblinding of an ongoing blinded clinical trial involving these participants. However, we have now delinked data for this study so that it may be shared upon publication as suggested.

In Data Availability, lines 746-748, we now state:

"Data for this study (doi:10.7303/syn10212437) will be made publicly available upon publication via Synapse, a software platform for open, reproducible data-driven science. The direct link to the data is: <https://www.synapse.org/#!/Synapse:syn10212437/files/>."

Comment 9: "Figure 4 - using grey font on white background reduces contrast. Panel B - font quite small on the x axis. Panels C/D - forced image view – more information would be conveyed if a table is presented. Provides no scale, font varied too much."

>>> We appreciate Reviewer 3's concern regarding font color and size in **Figure 4**. As suggested, we have changed font to black for panel A to improve visibility of the text, and have increased font size as possible to improve legibility in panel B, although we were unfortunately restricted by the length of the GO terms in some cases to increase font size any further. Detailed information for this plot, with GO terms, *P* values and enrichments are also provided in **Supplementary Table 5**, if additional clarification of the enriched terms is needed.

In direct response to Reviewer 3's comment on Figure 4 panels C and D, we provide detailed information underlying these plots in **new Supplementary Table 6**. We note that the varied box sizes and font carry information in these visualizations that we believe are helpful for the reader to see relationships between GO terms within a hierarchy. All GO terms presented have FDR < 0.05 as noted in the legend. The visualization allows us to collapse similar GO terms into categories such that each color represents a super-category with member boxes representing sub-categories and their size corresponding to the magnitude by which that sub-category contributes to the super-category. Thus, the visualization captures not only the magnitude of the significant GO terms (by box size) but also captures the subdivision and semantic relationships of the gene ontology structure (PMID: 21789182).

Comment 10: "Figure 5 provides an interesting view - but considering the overlap of nodes and edges - does not convey much information."

>>> We are sorry that Reviewer 3 feels this way, however, we believe that this figure conveys important information with respect to the levels of the causal network and the roles of the key drivers within it. We note that this style of figure has been used in other studies to organize and prioritize KDs (PMID 28017796).

Comment 11: "all results should be corrected for age, allergy severity and sex."

>>> We appreciate Reviewer 3's concern regarding these potential confounding factors. We carefully considered this and did not include these variables in our linear mixed-effects models for multiple reasons. First, regarding age and gender, we are not aware of any biological precedent that would indicate that these should have large effects on the expression of genes related to acute allergic reactions. In fact, when we assessed the effect these variables on the variance in gene expression across all genes and samples analyzed in our study using "variancePartition" (PMID: 27884101), we found that they had a relatively negligible effect, particularly when compared to the percent variance explained by inter-individual differences (see new **Supplementary Figure 6**).

In Methods, lines 629-632, we now state:

"Additionally, we used variancePartition,⁷⁵ to assess the contribution of technical and biological factors to variation in gene expression across the samples. This revealed minimal influence of age and gender (**Supplementary Figure 6**), and thus we did include these variables as covariates in the lme models next described."

Second, the design of the study with serial samples from the same subject over time and exposure (peanut vs. placebo) is such that each person serves as their own reference, as represented by the random effect of subject in the linear mixed effects models (see **Methods, lines 636-648**). Age and sex do not change within subject.

With respect to "correcting" for severity, we see this as problematic given that correction for severity would remove pertinent driving effects in the exact gene expression signatures we are seeking to observe. The current study is not designed or powered to address specific questions related to severity, and we argue that such questions should be explored in detail in future studies that are specifically designed for this purpose. We thank Reviewer 3 for this comment, as we find severity an interesting future direction as well.

Thank you for your thoughtful and kind consideration of our work for publication in *Nature Communications*.

Sincerely yours,

Supinda Bunyavanich, MD, MPH
Associate Professor
Icahn School of Medicine at Mount Sinai

Reviewer #1 (Remarks to the Author):

The authors addressed all of my concerns appropriately.

I have one minor comment: is the P value for CD22 (Table 2) in the replication cohort correct?

Reviewer #2 (Remarks to the Author):

The paper remains interesting and novel, and this time around easier to read. The addition of a replication dataset provides great reassurance around the robustness of the analyses and their interpretation.

I feel that my remarks have all been thoughtfully addressed.

Reviewer #3 (Remarks to the Author):

The authors have addressed most comments.

One remaining concern is the extent to which the second cohort validates the importance of the top 30 differentially expressed genes and the 6 driver genes. It would be good to know how many genes in the second cohort had P-values less than 0.05, and where the top 30 genes and the 6 driver genes rank in a list of genes sorted by P-values from cohort 2. The concern is that both the discovery cohort and the second cohort exhibit inflammatory responses, and many immune-related genes are expected to be differentially expressed in both groups - but the key genes characterizing an allergic response to peanuts may be different between the two cohorts. (i.e., any group of 30 immune-related genes might be enriched for low P-values in the second cohort). The top discovery cohort genes (top 30 & 6 driver) should be validated by testing whether their P-values in the second cohort are lower than those of most immune-related genes (maybe this could be tested with a Fisher's test or GSEA, using immune-related genes as background)

Reviewers' comments:

Reviewer #1:

Comment: “The authors addressed all of my concerns appropriately. I have one minor comment: is the P value for CD22 (Table 2) in the replication cohort correct?”

Response: Thank you for your positive feedback and critical eye. We have fixed this typo and also double checked the manuscript to ensure that there are no other typos.

Reviewer #2:

Comment: “The paper remains interesting and novel, and this time around easier to read. The addition of a replication dataset provides great reassurance around the robustness of the analyses and their interpretation.

I feel that my remarks have all been thoughtfully addressed.”

Response: We thank you for these positive comments.

Reviewer #3:

Comment: “The authors have addressed most comments.

One remaining concern is the extent to which the second cohort validates the importance of the top 30 differentially expressed genes and the 6 driver genes. It would be good to know how many genes in the second cohort had P-values less than 0.05, and where the top 30 genes and the 6 driver genes rank in a list of genes sorted by P-values from cohort 2. The concern is that both the discovery cohort and the second cohort exhibit inflammatory responses, and many immune-related genes are expected to be differentially expressed in both groups - but the key genes characterizing an allergic response to peanuts may be different between the two cohorts. (i.e., any group of 30 immune-related genes might be enriched for low P-values in the second cohort). The top discovery cohort genes (top 30 & 6 driver) should be validated by testing whether their P-values in the second cohort are lower than those of most immune-related genes (maybe this could be tested with a Fisher's test or GSEA, using immune-related genes as background)”

Response: Thank you for this feedback. For the revision, we had recruited a second cohort of peanut allergic children to replicate the findings we initially reported. Of the top 30 peanut genes identified in the discovery cohort, 28 exhibited statistically significant changes in expression during peanut challenge in the replication cohort (Table 2), representing a significant enrichment (OR=27.9, Fisher's exact test $P=9.5 \times 10^{-12}$) over that expected by chance. Further, the directions of effect for all 30 genes were consistent between the discovery and replication cohorts, lending further confidence. Such replication is remarkable and from our perspective, convincing. As replication studies of GWAS and other genetics studies have shown, one would not expect the

top 30 genes in a discovery cohort to be the top 30 genes in a replication cohort due to Type I errors and multiple testing.

To directly address Reviewer 3's concern about potential bias in the replication results due to common immune-related processes occurring in the discovery and replication cohorts, we performed additional analyses as suggested. Using transcriptome profiles from the replication cohort, we performed permutation testing with 1 million sets of 30 genes randomly selected from the Gene Ontology term 'immune system process' (N= 2952 genes; GO:0002376). None of these random selections of 30 immune-related genes overlapped with ≥ 28 genes significantly associated with peanut in the replication cohort (permuted $P= 9.9 \times 10^{-07}$) [PMID: 21044043]. This result provides reassurance that the significant replication was not due to broadly common immune processes.

Thank you for your thoughtful and kind consideration of our work for publication in *Nature Communications*.

Sincerely yours,

Supinda Bunyavanich, MD, MPH
Associate Professor
Icahn School of Medicine at Mount Sinai

REVIEWERS' COMMENTS:

Reviewer #3 (Remarks to the Author):

The authors have addressed all of my concerns.